# GenIR: Generative Visual Feedback for Mental Image Retrieval

Diji Yang[1][*]    Minghao Liu[1][3][*]    Chung-Hsiang Lo[2]    Yi Zhang[1]    James Davis[1]

[1]University of California Santa Cruz
[2]Northeastern University [3]Accenture
{dyang39, mliu40, yiz, davisje}@ucsc.edu
lo.chun@northeastern.edu
Project Page: https://visual-generative-ir.github.io

## Abstract

Vision-language models (VLMs) have shown strong performance on text-to-image retrieval benchmarks. However, bridging this success to real-world applications remains a challenge. In practice, human search behavior is rarely a one-shot action. Instead, it is often a multi-round process guided by clues in mind. That is, a mental image ranging from vague recollections to vivid mental representations of the target image. Motivated by this gap, we study the task of Mental Image Retrieval (MIR), which targets the realistic yet underexplored setting where users refine their search for a mentally envisioned image through multi-round interactions with an image search engine. Central to successful interactive retrieval is the capability of machines to provide users with clear, actionable feedback; however, existing methods rely on indirect or abstract verbal feedback, which can be ambiguous, misleading, or ineffective for users to refine the query. To overcome this, we propose GenIR, a generative multi-round retrieval paradigm leveraging diffusion-based image generation to explicitly reify the AI system's understanding at each round. These synthetic visual representations provide clear, interpretable feedback, enabling users to refine their queries intuitively and effectively. We further introduce a fully automated pipeline to generate a high-quality multi-round MIR dataset. Experimental results demonstrate that GenIR significantly outperforms existing interactive methods in the MIR scenario. This work establishes a new task with a dataset and an effective generative retrieval method, providing a foundation for future research in this direction [1].

## 1   Introduction

Recent Vision-language models (VLMs) have achieved decent results on standard text-to-image retrieval benchmarks [29, 13]. Despite this progress, transferring these capabilities into real-world applications remains challenging. One key limitation is that real human search behavior is often not one-shot or static; it unfolds through a sequence of actions, highlighting the necessity for interactive information retrieval (IIR) systems [24, 32, 1, 35]. Another limitation is that users frequently initiate a search to re-find the information they have seen before, which could be partial memory, vague clues, or vivid recall of the target images [3, 38, 5]. To address this scenario, we define Mental Image Retrieval (MIR) [2], where users iteratively refine their queries based on mental image (i.e., an image in mind) to retrieve an intended image from an image database.

---

[*]Equal contribution.
[1]Code and data are available at https://github.com/mikelmh025/generative_ir.
[2]The term is inspired by the Mental Image Reconstruction task [9] and shares the definition of "mental image", though their study purely from Neuroscience side which is different problem from ours.

39th Conference on Neural Information Processing Systems (NeurIPS 2025).

**Ours: Generative Mental Image Retrieval (GenIR)**

Query
Image Generator
Image to Image Retrieval
**Visual** Feedback

**Chat Mental Image Retrieval (ChatIR)**
Query
Text to Image Retrieval
Language Model
Verbal Feedback

**Interactive Mental Image Retrieval (PlugIR)**
Query
Text to Image Retrieval
Image Captioner
Verbal Feedback

Figure 1: Comparison of methods for Mental Image Retrieval task. **Top**: our generative method, which reifies the intermediate query using an image generator model and applies image-to-image search for retrieval. **Bottom**: Existing approach (ChatIR and PlugIR) which support multi-round query improvements based on verbal feedback.

Although MIR has not been explicitly formulated as a distinct task previously, as a subset of long-standing text-to-image IIR task [27, 5], some works have implicitly touched upon similar settings. ChatIR [12] positions a VLM (or ideally a human) as the active searcher. Seeing the initial query from the searcher, ChatIR uses Large Language Models (LLMs) to provide system feedback in question format to the user solely based on the textual dialogue history. The question is then answered by the searcher who has access to the ground-truth images (i.e., Mental Image). Next, the dialogue history will be appended with the question-answer pair and then be used as the search query. This implicitly positions ChatIR within the realm of MIR as a subset of interactive text-to-image retrieval where the human searcher holds the target image in memory. However, as shown in the bottom left of Figure 1, ChatIR has only verbal (text-based) feedback with no information from image space, resulting in generated question-answer pairs that may be redundant or irrelevant to the query refinement. PlugIR [11], as shown in the bottom right of Figure 1, advances this setup further by incorporating retrieval context—text captions of retrieved images into the query generation for subsequent rounds, aiming to produce more contextually relevant feedback and mitigate redundancy. Nevertheless, the major challenge exists, both methods remain constrained by significant limitations regarding feedback effectiveness. Even across multiple interaction rounds, these methods rely heavily on indirect, verbal feedback derived solely from retrieval failures. Such feedback is often abstract and interpretability-poor, providing users with little actionable insight or potentially misleading clues for refining subsequent queries. In vision-language embedding spaces, such as CLIP [23], minor textual edits can cause unpredictable changes in retrieval outcomes, making query refinement inherently a trial-and-error process. Consequently, the feedback offered by existing conversational retrieval approaches inadequately expresses the AI system's current understanding and fails to directly benefit users toward effective refinements. As ChatIR example shown in Figure 6, the verbal feedback for an image depicting a person wearing a motorcycle helmet: "a human is not wearing a hat". Although literally true, such feedback fails to capture the visual salience of the helmet and may steer the user's refinement toward irrelevant details, ultimately misleading the search process for images containing headwear.

Motivated by the need for more effective and interpretable system feedback, we propose GenIR, a generative interactive retrieval paradigm designed explicitly to provide clear, interpretable, and actionable visual feedback at each interaction turn. GenIR employs a straightforward but powerful iterative pipeline as shown in Figure 1: first, a text-to-image diffusion model generates a synthetic image from the user's current textual query; then, this synthetic image is used for retrieval from a database through image-to-image similarity matching. Crucially, the generated synthetic image serves as more than just a query, but acts as an explicit visualization of the system's internal understanding

(i.e., the representation of the query in the vision-language latent space), enabling users to clearly perceive discrepancies between their mental image and the system's interpretation so that can refine the query for the next round search.

Beyond its utility at inference time, GenIR also supports dataset construction for studying MIR. Following the common practice of using VLM to play as the human searcher [12, 11], we create an automated pipeline based on the GenIR framework. We present a multi-round dataset with each round consisting of a refined query, a generated synthetic image, and retrieved results with a correctness label. Our experiments demonstrate that GenIR outperforms existing MIR baselines, highlighting the significant advantage of using visual feedback over verbal feedback. Furthermore, our study indicates that our GenIR annotated query can result in better retrieval performance than annotation from ChatIR under the same retrieval setting. In summary, our contribution is as follows:

- Task: We formally define the Mental Image Retrieval (MIR) task, a subset of the multi-round interactive text-to-image retrieval task where the searcher has the target image in mind.

- Method: We propose a novel framework, GenIR, using the generative approach to provide intuitive, interpretable visual feedback revealing the optimization direction for user during multi-round query refinement.

- Dataset: We release a dataset with visually grounded feedback annotation, together with a curation pipeline which can support both MIR and general text-to-image retrieval tasks.

## 2 Related Works

### 2.1 Chat-based Image Retrieval

Conversational image retrieval, which use chat-like feedback to improve query for text-to-image IIR has gained attention as a way to improve search performance [19]. A foundational work, ChatIR [12], demonstrated improved retrieval accuracy through multi-round chats, where an LLM poses questions answered by a human with target image access. ChatIR also contributed a multi-round chat dataset and highlighted the utility of multi-round interaction for retrieval tasks. However, both the performance of their method and the quality of the curated dataset were limited by feedback efficiency issue (redundancy or misleading) as we discussed in Section 1. PlugIR [11] advanced this idea by proposing a plug-and-play image captioning model to collect feedback from retrieved images. This yielded context-aware and non-redundant verbal feedback [34]; however, both of these works and their related subsequent works [36, 37] remain fundamentally limited in their capacity to share nuanced visual representations with users.

In contrast, our work introduces a new modality into the loop: generated images that serve as visual hypotheses showing the system's understanding in the image space. Rather than relying on textual queries alone, our method synthesizes what the system "thinks" the user wants to search, enabling visual inspection and more precise system feedback.

### 2.2 Generative Image for Image Retrieval

Diffusion models have achieved great success in image reconstruction, that is, given a target image, a text encoder is trained to output human-readable language or a latent representation as input to the diffusion model, aiming to generate an image close to the target image [30, 31, 28]. However, applying such models directly to MIR is non-trivial, as human users can hardly provide actual images based on their mental images. A more feasible attempt is Imagine-and-Seek [15], which involves a one-time process that uses an image captioning model to generate a text description from the target image and then feeds it into a text-to-image diffusion model to generate a proxy image for retrieval. Yet, as discussed in the section 1, this single-round approach has been proven by multiple interactive retrieval works to be inferior in dealing with real-world applications [5, 27].

Apart from existing attempts, our approach uses image generation as a core step in the retrieval loop itself, not merely to improve retrieval performance for the current round, but to provide visual feedback to the user to potentially benefit the writing for the next round query. To our knowledge, this is the first work to integrate text-to-image generation into an interactive retrieval setting, enabling a closed-loop interaction that unifies generation, retrieval, and feedback within a single framework.

# 3 GenIR: Generative Retrieval with Visual Feedback

## 3.1 Task Formulation

We formally define the task of Mental Image Retrieval (MIR) as a subset of the text-to-image Interactive Information Retrieval [27]. MIR inherits the nature of multi-round interaction from IIR, while only focusing on the case where the user has an internal mental image that can not be directly accessed by the retrieval system. From an Information Retrieval theory perspective, MIR does not consider the Exploratory Search where a searcher has never seen the searching target [18, 32], but focuses on Known-item Search where the searcher has seen and can recall or partially recall the target information [33, 20, 2]. To define the task, we denote an image database $\mathcal{N}$, and let $I^{\text{target}}$ represent the image that the user has in mind. The retrieval task proceeds over multiple interaction rounds $t = 1, 2, ..., T$. At each round, the user formulates a textual query $q_t$ intended to approximate the mental representation of $I^{\text{target}}$. Based on the query $q_t$, the retrieval system returns a candidate image $I^{\text{retrieved}}_{(t)} \in \mathcal{N}$. Additionally, the system provides feedback signals to the user, potentially benefiting subsequent refinements of the query to bridge discrepancies between the retrieved candidate image and the user's mental image. The iterative process continues until the target image is successfully identified or a predefined maximum number of interaction rounds is reached.

## 3.2 Generative Retrieval Framework

This section details the GenIR framework and the rationale behind the design as shown in Figure 1.

**Query Formulation** At the beginning of each interaction round $t$, the human user formulates a textual query $q_t$ that represents their current visual intent. This query encapsulates the user's mental image description, which may evolve over subsequent rounds based on visual feedback provided. Users are encouraged to include both high-level descriptions (e.g., scene type, overall composition) and fine-grained attributes (e.g., color scheme, object details) to ensure comprehensive coverage of their mental image.

**Synthetic Image Generation** Central to the GenIR framework is the image generation component, which reifies textual queries into synthetic images. Specifically, given the user's query $q_t$, an image generator $G$ produces a synthetic visual representation $I^{\text{synthetic}}_{(t)} = G(q_t)$. This visual representation explicitly captures the retrieval system's interpretation of the query. Importantly, our framework is flexible and model-agnostic, allowing the use of various generative models (e.g., diffusion models, GANs, or any other generator). The key benefit of employing visual generation is that it significantly reduces ambiguity inherent in textual communication, offering an intuitive interface for users to identify discrepancies and refine their queries precisely.

**Image-to-Image Retrieval** With the synthetic image $I^{\text{synthetic}}_{(t)}$ generated, GenIR employs image-to-image retrieval as the core retrieval mechanism. Specifically, both synthetic and database images are embedded into a shared visual feature space using a suitable encoder (such as the image encoder from CLIP). Retrieval is then conducted by selecting the database image $I^{\text{retrieved}}_{(t)} \in \mathcal{N}$ that maximizes similarity to the synthetic image according to a visual similarity metric, commonly cosine similarity. Formally, $I^{\text{retrieved}}_{(t)} = \arg\max_{I \in \mathcal{N}} \text{cosine}\left(\phi(I^{\text{synthetic}}_{(t)}), \phi(I)\right)$, where $\phi$ denotes the image encoder. The use of image-to-image retrieval enhances retrieval quality by directly leveraging visual information, thus effectively bypassing limitations associated with purely textual queries.

**Feedback Loop** Upon viewing the generated synthetic image $I^{\text{synthetic}}_{(t)}$, the user gains valuable insight into the system's current interpretation of their query. This visualization allows the user to identify discrepancies between the generated image and their mental target, such as missing elements, incorrect attributes, or stylistic deviations. Based on this visual feedback, the user can then refine their query, guiding the system towards a more accurate retrieval result in the subsequent round. This iterative refinement loop continues until a stopping criterion is met, typically either a predefined maximum number of interaction rounds or the target is retrieved. By explicitly incorporating generative visualization as an intermediate step, GenIR makes the retrieval process interpretable, intuitive, and highly user-centric, thereby improving overall retrieval effectiveness in interactive retrieval setting.

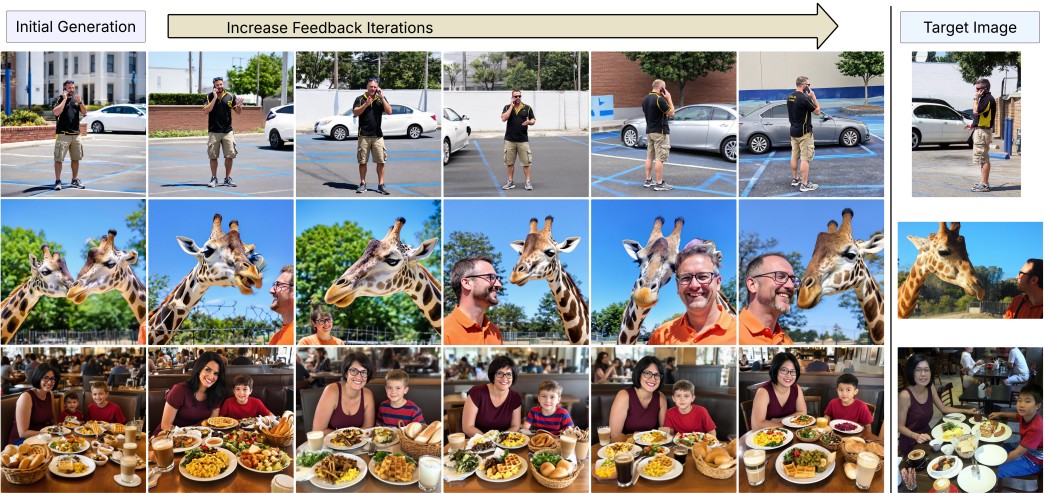

Figure 2: Visual progression of GenIR's image refinement process. Each row shows the evolution from initial generation (leftmost), through multiple feedback iterations (middle columns), to final generated result, alongside the target image (rightmost). Note how generated images progressively capture more accurate details with each iteration—improving clothing and posture (**row 1**), facial features and giraffe positioning (**row 2**), and dining scene composition (**row 3**).

### 3.3 Advantages of Visual Feedback in GenIR

GenIR's key advantage lies in its explicit visual feedback mechanism, addressing the limitations of existing systems that rely solely on ambiguous textual feedback. Text-based systems encode queries into a vision-language space, but this internal representation, the AI's interpretation or "visual belief", remains hidden from the user, making iterative refinement a challenging trial-and-error process.

GenIR alleviates this problem by using image generation to visualize the system's understanding of the textual query. This synthetic image serves as a direct projection of the query's meaning within the vision-language space into an interpretable visual form. Although the diffusion process does not provide additional information compared to using text to retrieve directly in the vision-language space, it reifies all the representations in an intuitive way. Consequently, users directly observe the model's internal visual belief (or "what the system thinks"), rather than navigating ambiguous textual interpretations, so as to intuitively identify discrepancies and refine their queries with knowledge of the details beyond text only.

Moreover, GenIR transitions the retrieval process from cross-modal matching (text-to-image) to same-modal matching (image-to-image). This allows subsequent search steps to leverage well-established visual similarity metrics that can capture spatial relationships and visual attributes that might be difficult to express precisely in text.

Figure 2 demonstrates this progression, showing how the generated images progressively improve with each iteration. Appendix A.3 contains a detailed version with the corresponding text queries that produced these refinements.

## 4 Experiment

### 4.1 Setting

**Task Definition**    Ideally, our approach would involve a human-in-the-loop. However, as a first step exploration along this direction, and considering the cost, we follow the standard setting of previous work [12, 11] to use a VLM to replace the individual who engages the mental image retrieval process. Specifically, we use a good-performing open-sourced VLM Gemma3 [25] to issue queries and improve the next round of queries based on the visual feedback provided by the image generator.

**Datasets**    We evaluate our method across four datasets with distinct visual domains to demonstrate the robustness of our approach. (1) **MS COCO** [16]'s 50k validation set, featuring common objects

in everyday contexts, provides a challenging testbed for retrieving images with complex scenes and multiple object interactions. (2) **FFHQ [8]**, comprising 70,000 high-quality facial portraits, represents the human-centric domain where fine-grained attributes (expressions, accessories, age) drive retrieval outcomes. (3) **Flickr30k** [21] contains 31,783 diverse real-world photographs showcasing people engaged in various activities across different environments. (4) **Clothing-ADC** [17], with over 1 million clothing images, introduces a specialized commercial domain extremely fine-grained subclasses (12,000 subclasses across 12 primary categories), enabling evaluation on highly specific attribute-based retrieval tasks. This domain diversity—spanning everyday objects, human faces, diverse activities, and fashion items—allows us to thoroughly evaluate how our generative retrieval approach performs across fundamentally different visual content types and retrieval challenges.

**Evaluation Metrics**    Following previous works in interactive image retrieval [12, 11, 5], we adopt Hits@$K$ as our primary evaluation metric, which measures the percentage of queries where the target image appears within the top-$K$ retrieved results. Specifically, we report Hits@10 to align with established benchmarks in the field. This metric effectively captures the practical utility of retrieval systems, as users typically examine only the top few results.

## 4.2    Implementation Details

We compare our proposed GenIR approach against several baselines to evaluate the effectiveness of generative visual feedback in Mental Image Retrieval:

**Verbal Feedback Methods (Baseline)**    We tested two Verbal-feedback baselines. The first one is ChatIR[12], which employs a human answerer for MSCOCO and ChatGPT to simulate a human for Flickr30k, with BLIP [14] serving as the questioner model. Second, we develop an enhanced version of ChatIR by replacing both sides with Gemma3 (in 4B or 12B parameter configurations), representing a stronger VLM-based baseline. Both methods operate without explicit visual feedback, relying solely on multi-round dialog for query refinement.

**Prediction Feedback (Baseline)**    This baseline incorporates visual feedback by showing the user (simulated by Gemma3) the top-1 retrieved image at each interaction round. The user examines this retrieved result and provides textual feedback describing discrepancies between the retrieved image and their mental target image. This approach represents a traditional interactive retrieval method [27] that leverages real images from the database but lacks the interpretability advantages of our generative approach.

**GenIR Configuration (Ours)**    GenIR provides explicit visual feedback through synthetic images generated from the user's textual query. To evaluate the sensitivity of our approach to generator quality, we test five state-of-the-art text-to-image diffusion models: Infinity [6], Lumina-Image-2.0 [22], Stable Diffusion 3.5 [4], FLUX.1 [10], and HiDream-I1 [26]. For all diffusion models, we use the default inference parameters as specified in their original works to ensure a fair comparison. Each model transforms the user's textual query into a synthetic image that visually represents the system's current understanding, which is then used for image-to-image retrieval through BLIP-2 [14].

## 4.3    Results and Analysis

**Performance on MSCOCO**    Figure 3 presents a comprehensive evaluation of our GenIR approach against traditional conversational retrieval baselines on the MSCOCO dataset, measured by Hits@10 percentage across increasing dialog lengths. All experiments were conducted using the full 50,000-image validation set as the search space, representing a challenging large-scale retrieval scenario.

The left graph demonstrates that our proposed GenIR method substantially outperforms all baselines, achieving approximately 90% retrieval accuracy even at the initial query and reaching nearly 98% by the tenth interaction round. This represents a significant improvement over the Prediction Feedback method (blue line), which reaches only  92% after ten rounds, and the Verbal Feedback baselines using Gemma3-12b (red line) and ChatIR (green line), which achieve  92% and  73% respectively. The substantial performance gap highlights the effectiveness of our visual feedback approach in providing clear, interpretable guidance for query refinement.

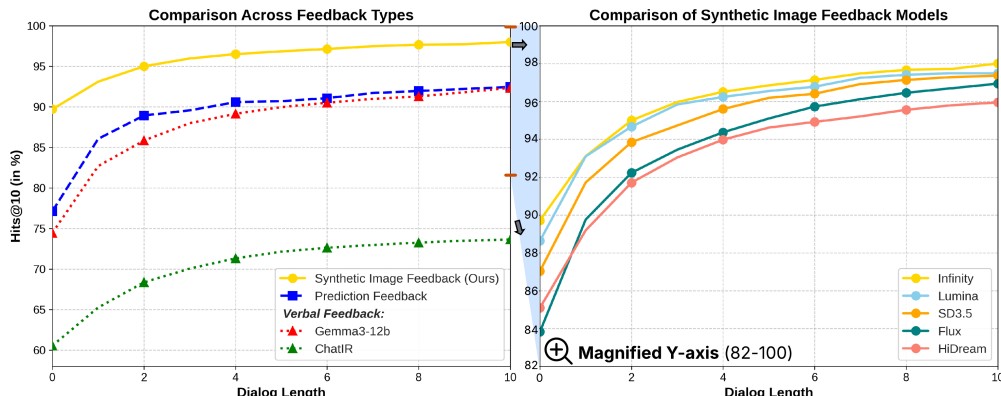

Figure 3: Performance Comparison on MSCOCO Dataset (Hits@10, 50k search space). **Left:** Our GenIR approach with Infinity diffusion model (Yellow) significantly outperforms all baselines, including Prediction Feedback (blue), Verbal Feedback with Gemma3-12b (red), and ChatIR (green). **Right:** Comparison of different text-to-image diffusion models within our GenIR framework, showing *consistent performance advantages across all generators*, with Infinity and Lumina achieving the best results after 10 interaction rounds.

The right graph further examines the impact of different text-to-image diffusion models on our method's performance. While all models demonstrate effective performance improvement over dialog rounds, Infinity and Lumina consistently outperform others, suggesting that higher-quality image generation contributes to more effective visual feedback. Notably, even with the lowest-performing generator (HiDream), our approach still achieves superior results compared to traditional feedback methods, demonstrating the robustness of our generative retrieval paradigm across different implementation choices.

To validate these quantitative findings with real users, we conducted a human evaluation study which found that 86% of the generated visual feedback was useful for query refinement; these human-annotated evaluations will be released alongside our dataset and code. Details of this study are provided in Appendix D.

**Cross-Domain Evaluation (FFHQ, Flickr30k, Clothing-ADC)** Figure 4 demonstrates GenIR's robust performance across three diverse visual domains. Our approach consistently outperforms all baselines regardless of domain characteristics, with particularly striking advantages in FFHQ (70% vs. 52% Hits@10 for the next best method) and ClothingADC (73% vs. 50%). Notably, ClothingADC represents an especially challenging scenario with over 1 million images in its search space—more than 20 times larger than the MSCOCO test set—yet GenIR maintains its substantial performance advantage. Even on Flickr30k, which shows higher baseline performance overall, GenIR maintains a clear 8-15% advantage throughout all interaction rounds. These results confirm GenIR's domain-agnostic effectiveness, especially with fine-grained visual details that text struggles to capture. Our consistent performance advantage across diverse domains and search space sizes demonstrates the approach's practical versatility.

**Effect of Vision-Language Model Size** Figure 5 examines the impact of VLM parameter scale (Gemma3-4b vs. Gemma3-12b) across different feedback methods on MSCOCO and FFHQ datasets. While larger models predictably deliver superior performance in all settings, the performance gap between model sizes is notably smaller with our Fake Image Feedback approach compared to alternative methods. Most significantly, our GenIR approach with the smaller 4b model consistently outperforms both Prediction Feedback and Verbal Feedback methods even when those methods utilize the larger 12b model. This finding demonstrates that visual feedback provides inherent advantages independent of model scale, enabling more efficient deployment without sacrificing retrieval quality.

**Prediction feedback is not always better than verbal** As shown in Figure 4, prediction feedback initially outperforms Verbal-feedback approaches but plateaus after 2-4 rounds, eventually being surpassed by text-only methods in longer dialogues. This suggests prediction feedback can trap the retrieval process in local minima, where iterative refinements based on a single retrieved image become

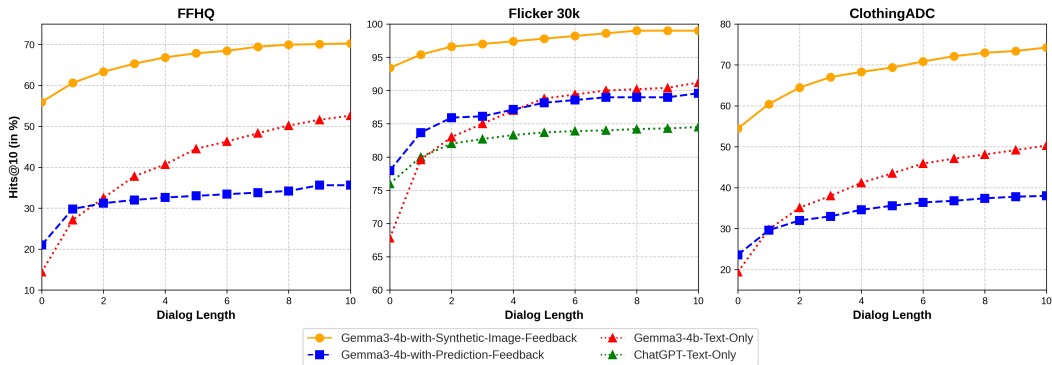

Figure 4: Performance Comparison on FFHQ, Flickr30k, and ClothingADC datasets (Hits@10). Our GenIR approach (yellow) consistently outperforms all baselines across domains, with particularly strong advantages in FFHQ and ClothingADC (the latter with a 1M+ image search space).

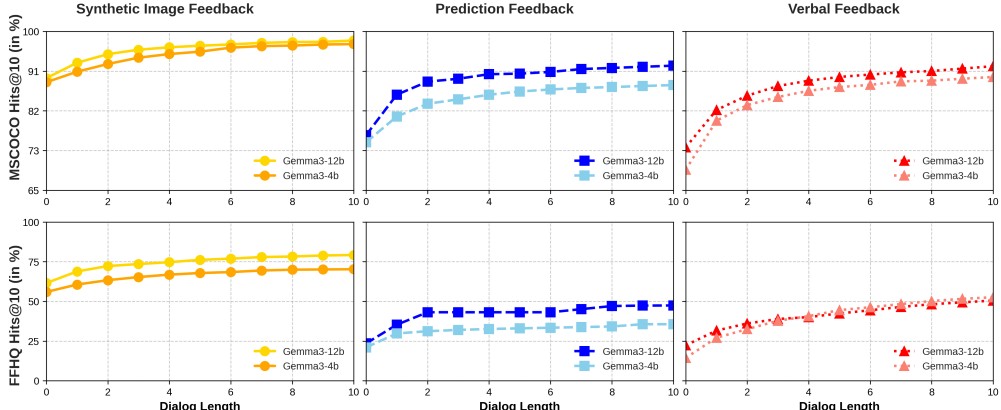

Figure 5: Analysis of vision-language model scale effects across feedback methods on MSCOCO (top) and FFHQ (bottom). While 12b models outperform 4b counterparts as expected, our GenIR with the smaller 4b model consistently surpasses alternative approaches even when using larger models.

increasingly incremental. In contrast, our generative approach provides a consistently improving trajectory by visualizing the system's understanding rather than showing database-constrained results.

**Generator-Agnostic Performance** Figure 3 (right) shows that performance differences between generators are minimal compared to the substantial gap between GenIR and baselines. This confirms our method's effectiveness derives from the visual feedback mechanism itself, not generation quality, enabling deployment with even simpler diffusion models.

## 5  Dataset Contribution

As a byproduct of our experimental framework, we construct a multi-round dataset for MIR task. Algorithm 1 describes our automated curation pipeline where the VLM formulates an initial query from the target image, then at each round: (1) a synthetic image is generated (Line 6); (2) the closest database image is retrieved (Line 7); (3) correctness is labeled (Line 8); and (4) the VLM refines the query based on visual discrepancies (Line 10). Unlike ChatIR, our dataset centers on visual feedback where both parties share understanding through images, reducing redundancy and misleading information.

Formally, the constructed dataset comprises a series of structured interaction rounds. At each round t, the data instance includes four key elements: a textual query $q_t$, a synthesized feedback image $I_t^{synthetic}$, a retrieved image from the database $I_t^{retrieved}$, and a binary label $y_t$ indicating whether the retrieved image correctly matches the target mental image. The dataset spans multiple domains

**Algorithm 1** Data Annotation Pipeline

---

1: **Notation:** $\mathcal{N}$: image pool
     $\mathcal{I}$: the set of ground-truth target images
     sim: Similarity search
2: Initialize dataset $\mathcal{D}$ as empty.
3: **for** each target image $I_{target} \in \mathcal{I}$ **do**
4:    $q_0 = \text{VLM}(I_{target})$                                            // Query formulation
5:    **for** $t = 1$ to $T$ **do**
6:        $I_t^{synthetic} = \text{Diffusion Generator}(q_t)$           // Synthetic Image Generation
7:        $I_t^{retrieved} = \arg\max_{x \in \mathcal{N}} \text{sim}(I_t^{synthetic}, x)$     // Image-to-Image retrieval
8:        $y_t \leftarrow \mathbb{I}[I_t^{retrieved} = I_{target}]$                // Assign correctness label
9:        $\mathcal{D}$ append $(q_t, I_t^{synthetic}, I_t^{retrieved}, y_t)$      // Record tuple in dataset
10:       $q_{t+1} = \text{VLM}(I_{target}, I_t^{synthetic})$     // Refine query based on visual feedback
11:    **end for**
12: **end for**
13: **return** dataset $\mathcal{D}$

---

(i.e., general, clothing, and human face), and each data point explicitly captures the shared visual grounding and query refinement trace. As a result, our dataset yields better query quality than ChatIR's, as experimentally validated in Appendix A.2. Furthermore, it uniquely provides a mid-step generated image for each retrieval round. It may serve as a testbed for studying MIR tasks and research problems such as visual feedback-driven retrieval and multi-round query refinement.

# 6   Limitation

Our study has two primary limitations: First, our VLM simulation assumes users have a clear, fixed target image in mind, whereas real users often begin with only partial or fuzzy mental representations. Second, our framework doesn't account for how mental images naturally evolve and clarify during the search process itself, as retrieval attempts often help users refine their own memory. Additionally, while visual feedback is useful in the majority of cases, generated images can sometimes mislead refinement through hallucinations or detail misalignments (detailed failure mode analysis in Appendix D.5). Future work should include human studies that capture these dynamic aspects of memory retrieval to validate GenIR's effectiveness in more naturalistic search scenarios. We leave more discussion on limitations and future work, including opportunities for RL-based system optimization, to Appendix E.

# 7   Conclusion

This paper introduced Mental Image Retrieval (MIR), a task modeling realistic interactive image search guided by users' internal mental images. Recognizing the limitations of verbal feedback, we proposed GenIR, a novel generative framework that employs an image generator to provide explicit and interpretable visual feedback. Notably, we expect GenIR to be a model-agnostic framework, allowing for the integration of various text-to-image generators (beyond diffusion models) and image-to-image retrieval models or algorithms. This plug-and-play capability enables leveraging any good-performing pre-trained models within the framework. Complementing the framework, we present an automated pipeline for curating a multi-round MIR dataset. Extensive experiments across diverse datasets demonstrate that GenIR significantly outperforms existing MIR approaches, highlighting the critical advantage of visual feedback for effective multi-round retrieval. Furthermore, evaluations under traditional text-to-image retrieval setting shows that queries refined by GenIR yield superior retrieval performance compared to those refined with purely verbal feedback (e.g., ChatIR), validating the quality and utility of our dataset for studying the general text-to-image retrieval task. This work provides a foundational step for future research into intuitive and interpretable interactive multimodal retrieval systems, encouraging further exploration of human-AI interaction dynamics and the role of generative vision models in enhancing interactive information retrieval.

## Acknowledgment

We thank Jinmeng Rao, Xi Yi, and Baochen Sun for helpful discussions and feedback on the early draft, and Linda Li for creating the visualizations of the system design.

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

# A  Additional results

## A.1  Limitations of Verbal-Only Feedback

Figure 6 illustrates a limitation of verbal-only feedback in interactive image retrieval. In this ChatIR example, when asked "is he wearing a hat?", the answer is "no," which is technically correct since the motorcyclist is wearing a helmet, not a hat. However, this verbal exchange could mislead the retrieval system by implying the person's head is uncovered, when in fact they are wearing protective headgear—an important visual attribute. Such semantic gaps in verbal feedback can lead to suboptimal query refinement.

Visual feedback approaches like GenIR potentially address this limitation by providing a synthetic image that would show the helmet, allowing users to immediately identify this discrepancy. This example demonstrates how visually grounded feedback can complement verbal descriptions by capturing visual details that might otherwise be lost in text-only exchanges, potentially leading to more accurate query refinement.

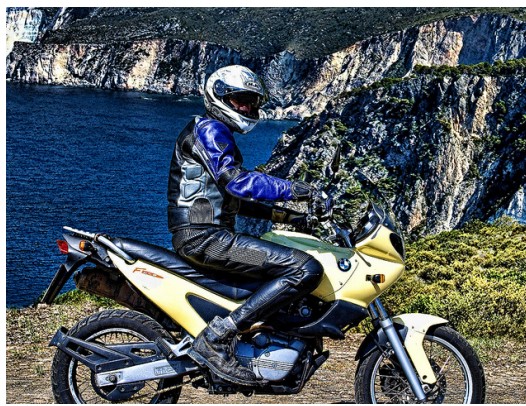

**Dialogue:**
```
"a man sits on a motorcycle next to
a very blue body of water"
"can you see any people?  just 1"
"is it a male or female?  male"
"i he facing the camera?  yes"
"is he happy?  don't know"
"is he wearing a hat?  no"
"what color is his hair?  can't see
hair"
"is it daytime?  yes"
"is it sunny?  yes"
"can you see the sky?  no"
"any animals?  no"
```

Figure 6: Example from ChatIR showing misleading verbal feedback. The highlighted question-answer pair demonstrates how verbal feedback can be technically correct but misleading—the person is not wearing a hat but is wearing a motorcycle helmet, a critical visual detail that verbal-only feedback fails to capture appropriately, potentially degrading retrieval performance.

## A.2  Comparative Analysis of GenIR Dataset Utility

To demonstrate the utility of our dataset, we conducted a comparative analysis between three different retrieval approaches: (1) **GenIR Synthetic Images** - using our generated synthetic images for image-to-image retrieval, (2) **GenIR Text** - using text queries generated through our GenIR framework for text-to-image retrieval, and (3) **ChatIR Text** - using verbal feedback-based text queries for text-to-image retrieval. Table 1 presents the Hits@10 performance on MSCOCO across dialog lengths. The results clearly demonstrate the superiority of visual feedback, with GenIR Synthetic Images achieving 89.71% even at initialization and 98.01% after 10 rounds. Notably, even the text queries generated through our GenIR framework significantly outperform ChatIR's verbal feedback approach (92.33% vs. 73.64% at round 10), confirming that the GenIR dataset contains higher-quality annotations that better capture user intent compared to purely text-based interactions.

## A.3  Visualization of Query and Image Refinement Process

The effectiveness of GenIR stems from its ability to provide explicit visual feedback that guides query refinement. Figure 7 illustrates this progressive refinement through multiple interaction rounds, highlighting how both textual queries and generated images evolve toward better alignment with the target mental image.

As shown in the figure, the initial queries tend to be verbose and contain extraneous details, resulting in generated images that capture the general scene composition but miss critical details or relationships. For example, in Round 0, the system generates an image showing two giraffes instead of the intended scene with one giraffe interacting with a person.

Table 1: Comparison of retrieval approaches across dialog lengths (Hits@10%)

| Dialog Length | ChatIR Text | GenIR Text | GenIR Synthetic Images |
|---|---|---|---|
| 0 | 60.56 | 74.47 | 89.71 |
| 1 | 65.26 | 79.09 | 93.11 |
| 2 | 68.36 | 83.41 | 95.00 |
| 3 | 70.06 | 85.69 | 95.97 |
| 4 | 71.32 | 87.63 | 96.51 |
| 5 | 72.14 | 88.79 | 96.85 |
| 6 | 72.63 | 89.71 | 97.14 |
| 7 | 72.97 | 90.39 | 97.48 |
| 8 | 73.26 | 90.98 | 97.67 |
| 9 | 73.50 | 91.75 | 97.72 |
| 10 | 73.64 | 92.33 | 98.01 |

Through subsequent rounds of feedback and refinement, the queries become increasingly precise and focused on the key visual elements that distinguish the target image. By Round 3, the query has been simplified and clarified to explicitly mention "a giraffe head near a man's face," resulting in a generated image that better captures the spatial relationship between the human and animal subjects.

By the final round (Round 10), the refinement process has successfully addressed the most important details—the giraffe's posture ("lowers its head towards him"), the man's appearance ("glasses and an orange shirt"), and the proper spatial arrangement. This progression demonstrates how GenIR's visual feedback mechanism enables users to identify discrepancies and iteratively align the system's representation with their mental image.

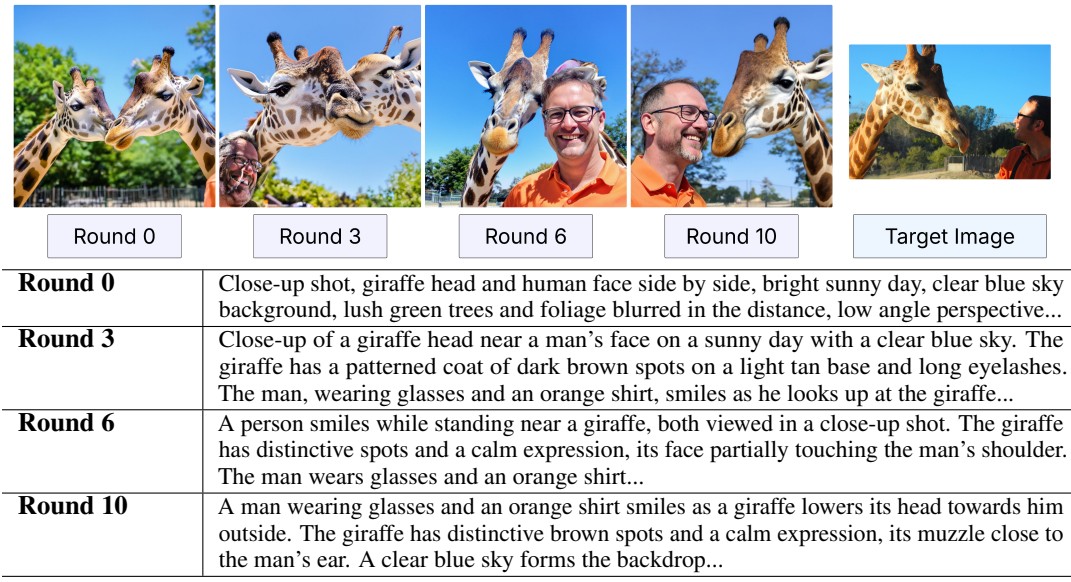

| Round 0 | Close-up shot, giraffe head and human face side by side, bright sunny day, clear blue sky background, lush green trees and foliage blurred in the distance, low angle perspective... |
|---|---|
| Round 3 | Close-up of a giraffe head near a man's face on a sunny day with a clear blue sky. The giraffe has a patterned coat of dark brown spots on a light tan base and long eyelashes. The man, wearing glasses and an orange shirt, smiles as he looks up at the giraffe... |
| Round 6 | A person smiles while standing near a giraffe, both viewed in a close-up shot. The giraffe has distinctive spots and a calm expression, its face partially touching the man's shoulder. The man wears glasses and an orange shirt... |
| Round 10 | A man wearing glasses and an orange shirt smiles as a giraffe lowers its head towards him outside. The giraffe has distinctive brown spots and a calm expression, its muzzle close to the man's ear. A clear blue sky forms the backdrop... |

Figure 7: Visual progression of GenIR's image refinement process across multiple rounds. The top row shows generated images evolving from initial generation (left) through intermediate rounds to final output (right), alongside the target image (far right). Below, the corresponding query texts show how descriptions become more precise and focused with each iteration. Note how the generated images progressively capture more accurate details—the spatial relationship between man and giraffe, facial features, lighting conditions, and background elements.

# B  Additional Experimental Details

## B.1  Hyperparameters

We provide the hyperparameters used for each of our experimental settings to ensure reproducibility:

### B.1.1  Diffusion Models Inference

Table 2: Hyperparameters for diffusion model inference

| Model | Inference Steps | Guidance Scale | Image Resolution |
|---|---|---|---|
| Infinity | N/A | 3.0 | $1024 \times 1024$ |
| Lumina-Image-2.0 | 50 | 4.0 | $1024 \times 1024$ |
| Stable Diffusion 3.5 | 28 | 3.5 | $1024 \times 1024$ |
| FLUX.1 | 5 | 3.5 | $1024 \times 1024$ |
| HiDream-I1-Fast | 16 | 0.0 | $1024 \times 1024$ |

### B.1.2  HiDream-I1 Model Adaptation

For our experiments with the HiDream-I1 model, we utilized a modified version compared to the original implementation available on GitHub. The standard HiDream-I1 model, which incorporates flow matching with dual CLIP encoders, T5, and Llama3.1-8b with 128 text tokens, requires approximately 55GB of VRAM for inference. Since our experiments were conducted on NVIDIA A6000 GPUs with 48GB VRAM, we employed the HiDream-I1-Fast variant with 4-bit quantization using the BitsAndBytesConfig approach. Following the implementation strategy by Hykilpikonna [7], we applied torch.bfloat16 precision and set low_cpu_mem_usage=True for all model components. This optimization reduced the memory footprint to under 30GB, enabling inference while maintaining reasonable generation quality. We observed that this quantized model preserved the essential characteristics needed for our visual feedback experiments, with minimal impact on the final retrieval performance.

### B.1.3  Image Retrieval Pipeline

For all image-to-image retrieval experiments, we used BLIP-2 with the following configuration:

- Feature dimension: 256
- Similarity metric: Cosine similarity
- Normalization: L2

### B.1.4  Vision-Language Models

For Gemma3 (both 4B and 12B variants), we used the following parameters:

- Temperature: 0.7
- Top-p: 0.9
- Max tokens: 500
- Repetition penalty: 1.1
- Sampling method: Greedy with temperature

## B.2  Compute Resources

All experiments were conducted using 4 NVIDIA A6000 GPUs with 48GB of VRAM each. The diffusion model inference for image generation was the most computationally intensive component of our pipeline, taking approximately 20 seconds per image generation at $1024 \times 1024$ resolution. The complete experimental suite, including all datasets and interaction rounds, required approximately 200 GPU hours to complete.

# C  Broader Impacts

## C.1  Potential Benefits

- **Enhanced User Experience**: By providing visual feedback, GenIR makes image retrieval more intuitive and aligns with how humans naturally think, potentially reducing frustration in search tasks.
- **Accessibility Improvements**: People who struggle with articulating precise textual queries (including those with language barriers or linguistic challenges) may find visually-guided search more accessible.
- **Creative Applications**: Artists, designers, and content creators could more effectively find visual references that match their mental concepts, enhancing creative workflows.
- **Educational Uses**: Teachers and students could more efficiently locate visual materials that align with conceptual understanding rather than relying solely on keyword matching.

## C.2  Potential Risks

- **Algorithmic Bias**: The diffusion models used for image generation may reproduce or amplify biases present in their training data, potentially leading to unfair representation in search results.
- **Computational Resource Requirements**: The use of generative models increases computational demands, which has both accessibility implications (requiring more powerful hardware) and environmental considerations (increased energy consumption).
- **Privacy Considerations**: As systems become better at representing users' mental images, questions arise about what information about user preferences and thinking might be inferred or stored.

Our approach aims to maximize the benefits while mitigating these potential risks through ongoing research and refinement of the methodology.

# D  Human Evaluation of Visual Feedback Utility

## D.1  Motivation

While our main experiments utilize VLM simulation (Gemma3) to evaluate the GenIR framework, we acknowledge that this approach cannot fully capture the nuanced ways humans form and refine mental images during search. VLM simulation, while effective for large-scale evaluation, may not accurately reflect how real users would interpret and benefit from visual feedback.

## D.2  Human Annotation Study

To address this limitation, we conducted a small-scale human evaluation study with the following methodology:

- **Dataset**: We selected 100 datapoints from our GenIR dataset. The evaluation primarily focused on comparing the ninth-round generated images with the actual target images.
- **Round Selection Rationale**: We specifically chose the ninth round because earlier rounds (1-6) typically captured broad image content but lacked significant details, while later rounds (7-9) produced more stable and detailed images. As the ninth round represents the final iteration in our framework, it provides the most refined visual representation for evaluation.
- **Annotation Task**: One human annotator evaluated each pair and classified whether the generated image was helpful for potential query refinement (binary classification: useful/not useful).
- **Evaluation Criteria**: The annotator assessed whether the synthetic image provided visual cues that would be valuable for further query refinement, focusing on whether the differences between generated and target images revealed actionable refinement opportunities.

## D.3 Results

Our human evaluation revealed that in 86% of cases, the synthetic images were judged as useful for query refinement. This strongly aligns with our quantitative results showing improved retrieval performance with visual feedback. Key observations include:

- Visual feedback was particularly helpful for refining fine-grained attributes (e.g., specific colors, textures, and spatial relationships) that are difficult to express precisely in text.

- In cases where the initial query was vague, the synthetic image helped clarify the system's interpretation, allowing for more targeted refinements.

- The instances where visual feedback was deemed unhelpful typically involved significant distortions or misinterpretations in the generated image that confused rather than clarified the search intent.

## D.4 Limitations and Future Work

This human evaluation, while informative, has several limitations:

- Limited scale (100 samples) and a single annotator

- The controlled setting differs from real-world search scenarios where users may have incomplete mental images

- The study does not capture the dynamic evolution of mental images during search

These limitations highlight the need for more comprehensive human-in-the-loop studies in future work. We plan to conduct larger-scale user studies with diverse participants to better understand how different user groups interact with and benefit from visual feedback in mental image retrieval tasks.

## D.5 Failure Mode Analysis of Visual Feedback

While our evaluation demonstrates that visual feedback is useful in 86% of cases (Section D), it is important to understand the failure modes where generated images can mislead the refinement process. We conducted a focused analysis of cases where GenIR failed to retrieve the target image, categorizing errors into three primary types based on how the generated visual feedback provided incorrect cues.

Table 3 presents our failure mode taxonomy with descriptions and representative examples from the MSCOCO dataset:

Table 3: Failure Mode Taxonomy: Categories of visual feedback errors that can mislead query refinement in GenIR

| Error Type | Description | Example (MSCOCO) |
|---|---|---|
| Limited Improvement | The generated image shows minimal visual change from the previous round despite query refinement, providing no new information to guide further refinement. | Image 000000004952: Generated images from rounds 8 and 9 are nearly identical, stalling the refinement process. |
| Hallucination Content | The generated image contains objects or scene elements that were not part of the user's intent, diverting the refinement process toward irrelevant details. | Image 000000020415: The 8th-round generated image hallucinates a shower head on the right side, an element entirely absent in the ground-truth, misguiding subsequent queries. |
| Retrieval-Detail Misalignment | The generated image misrepresents details that are crucial for retrieval but may be perceived as trivial by observers, creating a gap between visual feedback and retrieval objectives. | Image 000000026494: A long wooden bench in the ground-truth is incorrectly rendered as a single wooden chair. While seemingly minor, this detail is critical for accurate retrieval. |

**Analysis of Failure Patterns    Limited Improvement** (stagnation) typically occurs in later rounds (rounds 7-10) when the model has exhausted its capacity to refine the generation based on incremental textual changes. This suggests a limitation in the diffusion model's sensitivity to subtle query modifications.

**Hallucination Content** represents the most problematic failure mode, as it actively misleads users by introducing false visual elements. These hallucinations often stem from the diffusion model's tendency to complete scenes with common co-occurring objects, even when not specified in the query.

**Retrieval-Detail Misalignment** highlights a fundamental challenge: what appears visually acceptable to humans may miss crucial details that distinguish the target in the retrieval space. This suggests the need for retrieval-aware image generation that prioritizes discriminative details.

**Visualizations**    Figure 8 illustrates representative examples of each failure mode with visual comparisons between ground-truth targets and problematic generated images:

These visualizations demonstrate the current limitations of visual feedback generation and suggest future directions for improving feedback quality, such as retrieval-aware generation objectives or confidence-based hybrid feedback strategies.

## D.6    Computational Cost vs. Performance Analysis

While our GenIR approach demonstrates significant performance improvements over traditional feedback methods, it's important to consider these gains in relation to the computational overhead introduced by the generative process. This section provides a cost-benefit analysis of our approach compared to baseline methods.

### D.6.1    Comparative Computational Analysis

Table 4 presents a comparison of computational requirements between our GenIR approach and baseline methods. As expected, the integration of diffusion-based image generation introduces additional computational overhead compared to text-only methods like Verbal Feedback and text-based prediction feedback.

Table 4: Computational requirements comparison per interaction round

| Method | Compute Time (s) | Relative GPU Memory | Hits@10 at Round 5 |
|---|---|---|---|
| Verbal Feedback (Gemma3-12b) | 2 | 1.0× | 89.97% |
| Prediction Feedback | 2.5 | 1.2× | 90.70% |
| GenIR (Infinity) | 16 | 3.0× | 96.85% |
| GenIR (FLUX.1) | 12 | 2.5× | 95.10% |
| GenIR (Stable Diffusion 3.5) | 26 | 2.2× | 96.02% |
| GenIR (HiDream-FAST) | 17 | 2.1× | 94.62% |
| GenIR (Lumina-Image-2.0) | 27 | 1.3× | 96.55% |

Our analysis shows that GenIR with Infinity requires approximately 8 times more computation time per interaction round compared to the Verbal Feedback baseline. However, this computational investment yields a 6.9% absolute improvement in Hits@10 on the MSCOCO dataset by the fifth interaction round.

### D.6.2    Hybrid System: Balancing Performance and Efficiency

To address latency concerns while maximizing performance gains, we explored a hybrid approach that intelligently selects between fast Verbal Feedback and more powerful Visual Feedback. Our analysis shows that Visual Feedback uniquely succeeds in 22.3% of cases where Verbal Feedback fails, presenting an opportunity for strategic deployment.

Table 5 presents performance comparisons across different hybrid scenarios on MSCOCO:

- **Verbal Feedback**: Baseline using text-only feedback (fastest, 2s/round)

**Case 1: Limited Improvement (Image 04952)**

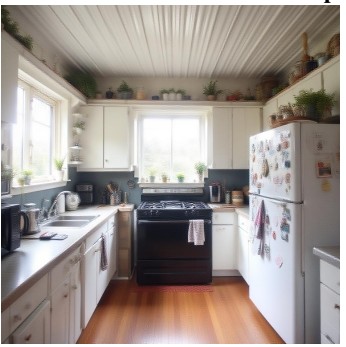 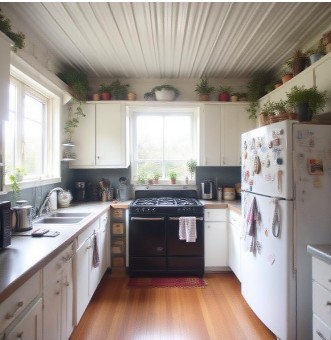

Round 8 Generated                    Round 9 Generated

**Case 2: Hallucination Content (Image 20415)**

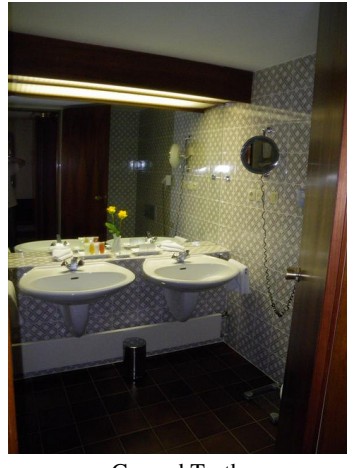 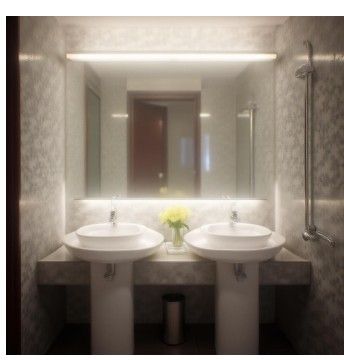

Ground Truth                    Round 8 Generated

**Case 3: Retrieval-Detail Misalignment (Image 26494)**

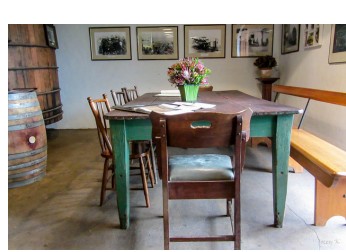 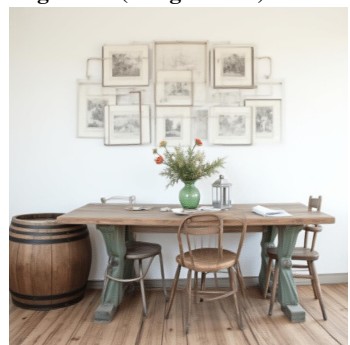

Ground Truth (bench)                    Round 9 Generated (chair)

Figure 8: Visual examples of GenIR failure modes. **Top**: Stagnation between rounds 8-9. **Middle**: Hallucinated shower head. **Bottom**: Bench misrepresented as chair.

- **Visual Feedback (Ours)**: GenIR with visual feedback (slower, 16s/round)

- **Hybrid Oracle**: Perfect selection of when to use Visual vs. Verbal feedback

- **Random Select**: Random selection using Visual Feedback for 22.3% of queries

Table 5: Hybrid System Performance: Comparison of Verbal Feedback, Visual Feedback, and Hybrid approaches (Hits@10%). The hybrid system uses Visual Feedback for 22.3% of queries and Verbal Feedback for 77.7%.

| Dialog Length | Verbal Feedback | Visual Feedback | Hybrid Oracle | Random Select | Verbal →Oracle | Verbal →Random |
|---|---|---|---|---|---|---|
| 0 | 74.48 | 89.71 | 93.40 | 77.88 | +18.92 | +3.40 |
| 1 | 82.73 | 93.11 | 95.97 | 85.06 | +13.25 | +2.33 |
| 2 | 85.83 | 95.00 | 97.43 | 87.89 | +11.60 | +2.06 |
| 3 | 87.97 | 95.97 | 97.91 | 89.76 | +9.95 | +1.79 |
| 4 | 89.13 | 96.51 | 98.20 | 90.81 | +9.07 | +1.68 |
| 5 | 89.96 | 96.85 | 98.30 | 91.50 | +8.35 | +1.54 |
| 6 | 90.49 | 97.14 | 98.59 | 91.98 | +8.10 | +1.49 |
| 7 | 90.98 | 97.48 | 98.64 | 92.42 | +7.67 | +1.44 |
| 8 | 91.27 | 97.67 | 98.79 | 92.69 | +7.52 | +1.42 |
| 9 | 91.80 | 97.72 | 98.79 | 93.13 | +6.99 | +1.33 |
| 10 | 92.33 | 98.01 | 98.88 | 93.62 | +6.55 | +1.29 |

Key findings from the hybrid system analysis:

- Even **random selection** yields meaningful gains (+3.40% absolute improvement at Round 0) with only +3 seconds average latency increase per query on MSCOCO.

- The **oracle hybrid system** demonstrates the upper bound of performance (+18.92% improvement at Round 0), showing significant room for improvement through intelligent query difficulty estimation.

- In difficult cases where verbal feedback requires many rounds or fails to converge, GenIR's ability to succeed in fewer rounds can actually be more time-efficient overall.

This analysis demonstrates that hybrid approaches offer a promising middle ground, providing substantial performance improvements with manageable latency overhead. Future work could develop learned policies to intelligently select between feedback modalities based on query characteristics.

### D.6.3 Optimizations and Efficiency Improvements

Several strategies can potentially reduce the computational overhead of our approach:

- **Model Distillation**: Smaller, distilled versions of diffusion models could reduce generation time with minimal performance degradation.

- **Early Stopping**: For many queries, acceptable performance can be achieved with fewer diffusion steps or feedbakc iterations.

- **Adaptive Generation**: Implementing a policy to skip generation in certain rounds where minimal refinement is expected could reduce overall computation.

### D.6.4 Real-World Deployment Considerations

The computational cost-benefit analysis varies significantly based on deployment context:

- **Interactive Search Applications**: The improved user experience and reduced number of interaction rounds may justify the additional per-round computation, especially since generation can be performed asynchronously while users review results.

- **Batch Processing**: For offline applications where multiple images need to be retrieved based on descriptions, the computational overhead may be prohibitive compared to traditional methods.

- **Specialized Domains**: In domains requiring high retrieval precision (e.g., medical imaging, satellite imagery), the performance improvements may justify computational costs regardless of application type.

Overall, while GenIR introduces non-trivial computational overhead, our analysis suggests that for many interactive retrieval scenarios, the performance gains and improved user experience justify the additional computational investment. Future work will focus on efficiency optimizations to further improve the performance-to-cost ratio of our approach.

# E   Limitation and Future Work

The current study lays the groundwork for Mental Image Retrieval (MIR) using generative visual feedback, and as such, its scope invites several avenues for future expansion:

**User Simulation for Initial Exploration**   Our use of Vision-Language Models (VLMs) to simulate user interaction is a deliberate methodological choice for this initial investigation of MIR. This approach aligns with common practice in pioneering new interactive AI tasks (e.g., as seen in prior interactive retrieval works [11, 12]), allowing for controlled, scalable, and reproducible exploration of the core GenIR framework. While this simulation provides a valuable starting point by assuming users have a relatively clear and fixed mental target, we recognize that real human users often begin with more partial or fuzzy mental representations. Future work should build upon our findings by conducting extensive human-in-the-loop studies. Such studies will be crucial for understanding how users with varying degrees of mental image clarity interact with GenIR and for refining the system to accommodate these more naturalistic scenarios.

**Dynamic Mental Image Evolution**   The dynamic evolution of a user's mental image during the search process, where the act of searching and receiving feedback can clarify or alter their internal representation, is a fascinating and complex aspect of human cognition. While our current work focuses on establishing the efficacy of generative visual feedback for a given mental target, investigating these interactive dynamics where the mental image itself co-evolves with system understanding was beyond the scope of this foundational study. We consider this a significant direction for future research. Exploring how GenIR can support or even leverage this iterative refinement of the user's own memory presents a rich area for subsequent investigation.

**Reinforcement Learning-Based System Optimization**   While our current plug-and-play design intentionally provides flexibility for researchers to leverage state-of-the-art or domain-customized models, end-to-end optimization of the GenIR framework presents exciting opportunities for future work. We envision optimization from two complementary perspectives: the framework architecture and the unique dataset it produces.

**GenIR as a Multi-Agent System.** At its core, GenIR functions as a multi-agent system where the generator and retriever collaborate to solve retrieval tasks. By introducing explicit visual feedback into this loop, we create an interactive paradigm potentially amenable to established multi-agent optimization techniques. The modular design makes GenIR a flexible testbed where researchers can investigate how different agent capabilities impact system dynamics and optimization potential.

**Trajectory-Rich Dataset for RL.** Our dataset provides unique opportunities for system optimization through its rich, multi-step interaction trajectories:

- **Warm-up Training**: Successful ($query \rightarrow generated\ image \rightarrow retrieval\ result$) trajectories can provide initial supervised training before RL fine-tuning. This warm-up stage, standard practice for improving RL stability and sample efficiency, is particularly crucial for navigating the vast action space inherent to large generative models.

- **Trajectory-Based Optimization**: Full interaction histories enable advanced trajectory-based optimization through process supervision (e.g., training reward models) rather than relying solely on delayed final rewards. This allows the system to learn from intermediate feedback quality, not just final retrieval success.

- **Diagnostic Analysis**: Intermediate generated images provide unprecedented insight into how and why generative models misunderstand queries in task-oriented settings. This offers direct, actionable guidance for system-level debugging and task-specific optimization.

**Optimization Strategy.** We envision freezing the retriever while optimizing the image generator end-to-end. This design choice offers flexibility—the retriever need not be a trainable deep learning model, allowing use of fixed systems like Google Search or domain-specific rule-based search. For the generator, successful optimization depends on two objectives: (1) correctly reflecting text queries visually, and (2) distinguishing between retrieval targets and incorrect candidates in the image space, ensuring seamless component collaboration.

These outlined possibilities represent starting points for engaging the research community in multi-agent system optimization. We believe the community will discover many additional opportunities building on this foundation. Our work provides essential building blocks—a flexible framework and trajectory-rich dataset—to unlock the full potential of generative visual feedback in interactive AI tasks.

