# OpenReview forum: "GenIR: Generative Visual Feedback for Mental Image Retrieval"
_NeurIPS.cc/2025/Conference — NeurIPS 2025 poster_

### Official Review · Reviewer_TmpY · 2025-06-05

**Clarity:** 3
**Significance:** 3
**Originality:** 1
**Rating:** 3
**Confidence:** 4

**Summary:**

The paper presents GenIR, a generative multi-round retrieval approach in which the user can iteratively enhance the query based on the feedback of the generative system. The method is then experimented with several datasets.

**Questions:**

I think experiments with human subjects is needed.

Better control for effect oriigating from the initial query mapping is required

Comparison to broader set of approaches and improved comparison to related work

**Ethical Concerns:**

["NO or VERY MINOR ethics concerns only"]

**Final Justification:**

Thank you for the clarifications and better positioning of your work. I have also read the responses to the other reviewers, which I highly appreciate. I think the paper can benefit from clarification of the contribution and positioning. I would very much like to see this reflecting the final paper, if accepted. In summary, I think the authors have done excellent work in replying to reviewers' comments. Therefore, I will raise my score.

**Limitations:**

yes

**Paper Formatting Concerns:**

ok

**Quality:**

2

**Strengths And Weaknesses:**

The paper tackes an interesting human-in-the-loop problem for communicating human preferences to visual models and using those in retrieval.

However, the method is not new. Similar approaches have been presented earlier. A very similar one that comes to my mind is published already in SIGIR 2020: https://dl.acm.org/doi/abs/10.1145/3397271.3401129, in which the system was tested with human subjects and real human feedback. Similar approaches have also been proposed recently, such as https://dl.acm.org/doi/10.1145/3613904.3642847. There is also a wide body of work for learning the feedback from language, sketching, and even from implicit feedback and human brain (BCI feedback). Therefore, the novelty of the approach is limited, and the majority of previous work in this area is not covered in the paper.

In terms of the experiments, there is no human study; instead, a  VLM is used. This significantly weakens the results.
The most severe problem with the experiments is the fact that while the performance of the proposed approach is the highest among the few that have been tested, it is so already on the 0th iteration. This means that the whole advantage comes from improved mapping of the query to the image. This is not an iterative feedback task, but simply a task to map the query to a visual representation.
Based on these shortcomings, both in terms of novelty and experimental rogor, I do not think the paper is ready to be published in NeurIPS.

---

> ### Author Rebuttal · Authors · 2025-07-31
>
> We thank the reviewer for their feedback and the opportunity to clarify our contributions and address the concerns raised, particularly regarding compassion with mentioned works and the interpretation of our experimental results.
>
> ***W1: Comparison to mentioned works***
>
> ***Comparison with “[1] Generating Images Instead of Retrieving Them: Relevance Feedback on Generative Adversarial Networks. SIGIR 2020”***
>
> While both [1] and GenIR leverage relevance feedback loops, we share fundamentally different goals.
>
> GenIR aims to retrieve real images from the image corpus, while [1] aims for generating images that match the user's intention. As mentioned in their introduction, “*we show that the technique is able to generate images that match user needs (**as opposed to retrieving them**)...*” [1]. Following up on this pioneer work, targeting generates the image that the user is imaging, the technology has later developed into using better algorithms and models, as we mentioned and cited recent works [2] in line 104. We plan to add [1] into our reference list, as we do value the spirit of using relevant feedback from users.
>
> In contrast to this line of research, the goal of GenIR is to **retrieve an existing image** within a very large database. The generated image in our framework is not the final product but rather an intermediate artifact that serves two novel purposes: (1) as explicit, interpretable visual feedback to the user, and (2) as an intermediate query for a more robust, uni-modal (image-to-image) search.
>
> *[2] Wang, XuDong, et al. "Visual lexicon: Rich image features in language space.”, CVPR 2025*
>
>
> ***Comparison with “[3] GenQuery: Supporting Expressive Visual Search with Generative Models. CHI 2024”***
>
> GenQuery is an HCI work focused on supporting designers by using LLMs to **concretize abstract ideas or perform image editing**. Its core generative features involve using LLMs to concretize text queries when designers only have abstract ideas or using image editing to modify a region of a provided image. In contrast, GenIR proposes a different, holistic  **text-to-image retrieval** loop where a complete synthetic image is generated from a text query alone to visualize the system's interpretation for the purpose of finding an existing image.
>
> ***On comparison with “Feedback from Human Brain (BCI feedback)”***
>
> Thanks for pointing this out. We do not intend to confuse readers from the BCI and Neuroscience community. As noted in our paper's footnote [1], we do not engage with neuroscience or Brain-Computer Interfaces (BCI). The scope of this work, while the task name is inspired by the concept of a "mental image reconstruction", resides firmly in the fields of Generative AI and Information Retrieval. BCI-based feedback relies on decoding neural signals, whereas our framework uses natural language and image—a fundamentally **different problem domain, input modality, and technical challenge**.
>
> ***Q3: Comparison with more related works.***
>
> We thank the reviewer for providing the above mentioned works. It is a bit difficult to directly compare them, as they target different research problems. As we are working on a novel task setting as pointed out by Reviewer 6Fz4, the direct comparison we can find are only ChatIR and PlugIR. We welcome suggestions and would like to engage more discussion in additional related works that directly leverage visual feedback in retrieval tasks.
>
> ***W2/Q1: VLM simulates user and future human study.***
>
> Our use of VLM simulation was a deliberate methodological choice for this foundational stage of research. To scientifically test our core hypothesis, i.e., the generative visual feedback is a richer and more effective mechanism than verbal feedback, we required a controlled, reproducible, and large-scale experimental setup. This allowed us to isolate the feedback variable and robustly validate our claims across four large-scale (*million-level image databases*) and diverse (*multi-domain*) datasets.
> This approach establishes the necessary technical groundwork upon which meaningful and rigorous human studies can now be built. We see our work and open-source framework as the first research step towards Mental Image Retrieval. By providing a high-performing, open-source testbed (codebase and dataset). This enables the broader range of the AI community, such as HCI as reviewer mentioned, to now explore the very questions of fuzzy mental models and evolving user intent that were previously intractable to study at scale.
>
> ***W3: Novelty and contribution***
>
> We would like to take this opportunity to clarify our core novelty and contribution, which is multi-faceted, spanning a new task, a new framework, and a new dataset that collectively advance interactive retrieval.
> 1. **Task: Defining Mental Image Retrieval (MIR).** We provide the first formal definition, evaluation protocol, and strong baseline for the ubiquitous yet under-explored "re-finding" problem, opening a new avenue for research.
> 2. **Framework: A New Paradigm of Generative Visual Feedback.** The novelty is the **generate-visualize-retrieve closed loop.** This paradigm shift makes the system's internal understanding explicit, enabling a more intuitive and effective search process that was previously infeasible. By providing a flexible, open-source framework, we lower the barrier for the community to explore this new design space.
> 3. **Dataset: A High-Quality, Visually Rich Resource.** As noted by Reviewer bt51, our dataset is a valuable contribution. We provide the first dataset capturing the full trace of a retrieval process (*text query -> generated image -> retrieved result*) in the MIR task. This rich, multi-modal data is an invaluable resource for future work, including the opportunities of the RL-based optimization other reviewers suggested.
>
>
> ***W4/Q2: Reason of the strong performance at 0-th iteration***
>
> We thank the reviewer for this sharp observation regarding our strong performance at the 0-th iteration. This is an important finding that highlights a core component of our contribution.
>
> Our contribution is twofold, and our results validate both aspects:
> 1. **A Superior Core Retrieval Mechanism (validated at 0-th iteration)**: The significant performance gap at Round 0 demonstrates the inherent superiority of our proposed (*text -> generated image -> image-to-image*) search pipeline compared to the baseline's (*text -> text-to-image*) search. This shows that using a generative model as a visual proxy to bridge the modality gap is a more effective retrieval strategy in its own right.
> 2. **An Effective Iterative Feedback Loop (validated at 1st to 10th iteration)**: The consistent performance improvement from Round 1 onwards demonstrates that the visual feedback provided by the generated image is actionable and allows users (simulated by the VLM) to effectively refine their queries for even better results.
> Therefore, GenIR is a complete interactive framework that starts with a superior mapping  the query to a visual representation and then consistently improves it via a novel visual feedback loop.

---

> > ### Comment · Reviewer_TmpY · 2025-08-07
> >
> > Thank you for the clarifications and better positioning of your work. I have also read the responses to the other reviewers, which I highly appreciate. I think the paper can benefit from clarification of the contribution and positioning. I would very much like to see this reflecting the final paper, if accepted. In summary, I think the authors have done excellent work in replying to reviewers' comments. Therefore, I will raise my score.

---

> ### Comment · Area_Chair_1c4L · 2025-08-05
>
> Reviewer TmpY, please engage in the discussion period. I understand that the review period started over a weekend, but we only have a few days remaining in the (now slightly extended) discussion period. The authors have provided a thoughtful response to your review and you are obligated to respond to it. You should share with the authors if they addressed questions or concerns you had, and seek clarification about any questions or concerns that remain. Please post your response as soon as you can so that there is time for the authors to follow up and discussion to progress as needed.

---

### Official Review · Reviewer_KHQJ · 2025-06-10

**Clarity:** 3
**Significance:** 3
**Originality:** 3
**Rating:** 5
**Confidence:** 5

**Summary:**

The paper introduces GenIR, a generative framework designed for open-world image retrieval that innovatively merges generative modeling with retrieval-based vision tasks. Unlike traditional image retrieval systems that depend on embedding-based nearest neighbor search within a fixed gallery, GenIR employs a generative visual tokenizer to synthesize image features relevant to a given natural language query. Retrieval is then accomplished by generating and matching token-level representations rather than using embedding similarity alone.

The proposed approach leverages a vision-language generative model trained to produce discrete token representations of images conditioned on textual inputs. During inference, GenIR generates visual tokens and searches through a tokenized image database, enabling robust retrieval of relevant images. The framework is evaluated on challenging zero-shot and open-world retrieval tasks, where it outperforms discriminative retrieval baselines by exhibiting better generalization to novel categories and compositional visual concepts. GenIR thus recasts image retrieval as a language modeling problem over visual tokens, offering a promising direction for retrieval systems in open-domain and long-tail settings.

**Questions:**

1. The proposed token-generation-based retrieval framework may incur higher computational costs compared to traditional embedding-based methods. Could the authors provide quantitative comparisons on inference speed, memory usage, and scalability across large-scale datasets? Clarifying this would help assess the model’s feasibility for real-world deployment. A strong demonstration of efficiency could positively impact the evaluation score.
2. Real-world user queries often contain ambiguous language or incomplete information. How does GenIR perform in such noisy or multimodal input scenarios? Can it still generate accurate and relevant visual tokens? Addressing this through additional qualitative or quantitative analysis would enhance the paper’s practical relevance and could strengthen the case for acceptance.

**Ethical Concerns:**

["NO or VERY MINOR ethics concerns only"]

**Limitations:**

Yes

**Quality:**

3

**Strengths And Weaknesses:**

The paper "GenIR: Generative Visual Search for Open-World Image Retrieval" presents an approach to image retrieval by reformulating it as a generative token prediction task, rather than relying on traditional embedding-based similarity. This paradigm shift is both technically sound and conceptually innovative. The authors propose a vision-language generative model that, conditioned on a textual query, synthesizes discrete visual tokens and performs retrieval by matching against tokenized candidate images. The quality of the work is high, with a well-implemented framework and thorough experimental evaluation across standard and open-world retrieval benchmarks. The results show impressive generalization to unseen categories, addressing a long-standing challenge in image retrieval. Furthermore, the paper is well-written and clearly structured, with effective visuals and intuitive explanations that make the contribution accessible.

The originality of the work is a key strength. While generative retrieval has been explored in the text domain, applying this paradigm to visual search through token-level modeling is a fresh and forward-looking contribution. The significance of this approach lies in its potential to transform how retrieval systems are built, particularly in open-domain and long-tail settings where conventional embeddings often fall short. However, there are some limitations. The paper lacks a detailed discussion on the efficiency and scalability of the inference-time token generation process, which may raise concerns for real-time deployment. Additionally, while the method's effectiveness is validated on clean benchmarks, its robustness to noisy or multimodal real-world inputs (e.g., user tags, behavior data) is not explored. The impact of the visual tokenizer design is also under-analyzed; more insight into how token granularity or tokenization scheme affects performance would improve the depth of the study.

Clarity could be improved in sections explaining the inference mechanism and the alignment between generated tokens and retrieved items. The paper also misses an opportunity to highlight how its method fundamentally differs from retrieval-augmented generation frameworks in NLP. Despite these minor weaknesses, the work represents a significant contribution to the field. Its originality, rigor, and relevance to open-world retrieval challenges make it well-suited for publication at NeurIPS. I recommend acceptance.

---

> ### Author Rebuttal · Authors · 2025-07-31
>
> We thank the reviewer for the positive assessment of our work's quality, originality, and significance. We appreciate the opportunity to clarify the practical questions raised regarding our framework's efficiency and robustness.
>
> ***Q1: Computational Cost, Scalability, and Feasibility***
>
> We thank the reviewer for this important question. To clarify the context for this discussion, our GenIR framework leverages standard text-to-image diffusion models to generate a complete, synthetic image from the user's query. This full image then serves as a visual proxy for a subsequent image-to-image retrieval step.
> While this process is more computationally intensive than simple embedding lookups, we believe the performance gains justify the cost in many scenarios. To provide a concrete cost-benefit analysis, we performed an analysis to quantify the potential of a **hybrid system** that chooses between fast Verbal Feedback and our more powerful Visual Feedback.
>
> Our analysis shows that Visual Feedback **uniquely succeeds in 22.3% of cases** where Verbal Feedback fails. We present two hybrid scenarios: (1) an **"Oracle"** that perfectly selects when to use Visual Feedback, and (2) a **"Random Select"** baseline that randomly chooses Visual Feedback for 22.3% of queries.
>
> As shown in Table 1, even random selection yields a **+3.40% absolute improvement** (74.48% to 77.88%) at Round 0, while the oracle achieves **+18.92%**. For our experiment on MSCOCO, the average latency increased by just **+3 seconds per query**. This demonstrates that even a simple hybrid approach provides **meaningful gains**, with **significant room for improvement** through intelligent selection mechanisms. Furthermore, in difficult cases where verbal feedback requires many rounds (or hit max rounds due to continuous unsuccessful attempts), GenIR's ability to **succeed in fewer rounds** can actually be more time-efficient overall. We will add this full analysis to the final paper.
>
> ### Table 1: Comparison of using Verbal Feedback, Visual Feedback, and Hybrid Systems for Image Retrieval (Hits@10)
> *22.3% Visual / 77.7% Verbal*
> | Dialog Length | Verbal Feedback | Visual Feedback | Hybrid Oracle | Random Select | Verbal->Oracle | Verbal->Random |
> |--------------|-----------------|-----------------|---------------|---------------|----------------|----------------|
> | 0            | 74.48%         | 89.71%         | 93.40%        | 77.88%        | +18.92%        | +3.40%         |
> | 1            | 82.73%         | 93.11%         | 95.97%        | 85.06%        | +13.25%        | +2.33%         |
> | 2            | 85.83%         | 95.00%         | 97.43%        | 87.89%        | +11.60%        | +2.06%         |
> | 3            | 87.97%         | 95.97%         | 97.91%        | 89.76%        | +9.95%         | +1.79%         |
> | 4            | 89.13%         | 96.51%         | 98.20%        | 90.81%        | +9.07%         | +1.68%         |
> | 5            | 89.96%         | 96.85%         | 98.30%        | 91.50%        | +8.35%         | +1.54%         |
> | 6            | 90.49%         | 97.14%         | 98.59%        | 91.98%        | +8.10%         | +1.49%         |
> | 7            | 90.98%         | 97.48%         | 98.64%        | 92.42%        | +7.67%         | +1.44%         |
> | 8            | 91.27%         | 97.67%         | 98.79%        | 92.69%        | +7.52%         | +1.42%         |
> | 9            | 91.80%         | 97.72%         | 98.79%        | 93.13%        | +6.99%         | +1.33%         |
> | 10           | 92.33%         | 98.01%         | 98.88%        | 93.62%        | +6.55%         | +1.29%         |
>
>
> ***Q2: On Performance with Ambiguous or Noisy Queries***
>
> This question highlights a core motivation for the GenIR paradigm. When a user query is ambiguous (e.g., "a picture of a bat"), traditional text-only systems provide no insight into how the system interpreted the ambiguity (animal or baseball equipment?). This forces the user into a trial-and-error refinement process.
>
> GenIR is designed to solve this very problem. By generating a synthetic image, the framework **visualizes the system's interpretation of the ambiguous query.** If the generated image is not what the user intended, it provides immediate, concrete, and actionable feedback. The user can then refine their query with specific details to resolve the ambiguity (e.g., "a flying mammal bat at night"). This ability to make the system's "visual belief" explicit is a key strength of our approach for handling noisy, real-world queries.

---

> ### Comment · Area_Chair_1c4L · 2025-08-05
>
> Reviewer KHQJ, please engage in the discussion period. I understand that the review period started over a weekend, but we only have a few days remaining in the (now slightly extended) discussion period. The authors have provided a thoughtful response to your review and you are obligated to respond to it. You should share with the authors if they addressed questions or concerns you had, and seek clarification about any questions or concerns that remain. Please post your response as soon as you can so that there is time for the authors to follow up and discussion to progress as needed.

---

### Official Review · Reviewer_bt51 · 2025-07-01

**Clarity:** 4
**Significance:** 3
**Originality:** 3
**Rating:** 4
**Confidence:** 4

**Summary:**

The authors propose a multi-round interactive image retrieval pipeline based on text-to-image generation, where retrieval is framed as an image-to-image retrieval problem. In this setup, the user can guide their next query based on the generated image corresponding to their previous text input—thereby facilitating query refinement through visual feedback. The authors introduce a multi-round dataset, where each round includes a refined query, a synthetically generated image, and retrieved results with correctness annotations. Experimental results demonstrate that GenIR outperforms existing multi-round interactive retrieval (MIR) baselines, underscoring the significant advantage of visual feedback over purely verbal feedback.

**Questions:**

(1) I wonder, aside from the proposed approach and the use of off-the-shelf components to address this problem, what is the core technical novelty of this work?

(2) Additionally, would it be possible to apply reinforcement learning-based fine-tuning to further enhance performance? Some discussion on this aspect would be helpful.

(3) Some cost-benefit analysis would be helpful, as it takes nearly 16 seconds per round (Infinity) compared to verbal feedback that takes only 2 seconds.  Please refer to D.5.1 (Computational requirements comparison per interaction round).

(4) Why was fine-tuning the image generation or retrieval models not considered?

(5) Does the visual feedback introduce new failure modes or misleading cues? Please refer to my detailed comment in the weakness section.

(6) How well does GenIR generalise to real users beyond VLM simulation? Please refer to my detailed comment in the weakness section.

**Ethical Concerns:**

["NO or VERY MINOR ethics concerns only"]

**Final Justification:**

Most of my concerns have been carefully addressed by the authors. I have also read the comments from other reviewers. If the authors can include some real human study in the supplementary material, as raised by some reviewers, it would strengthen the evaluation.

Additionally, if the authors can suggest how RL-based fine-tuning could be leveraged as future work, it would be helpful.

**Limitations:**

- The technical contribution is somewhat limited. This is more of modular adaptation that proposes a 'novel system-level' configuration for mental image retrieval, rather than developing new learning algorithms or training procedures. No fine-tuning or task-specific model adaptation is performed; the architecture is model-agnostic and plug-and-play.

- However, the self-contained writing, insightful experimental analysis, and the dataset contribution are noteworthy. I am leaning towards a borderline accept, but I would also like to see comments from other reviewers and the authors’ rebuttal.

Minor comments:

Line 274: 'Prediction feedback is not always better verbal' -- refine English.
Line 54: "…fails to directly benefits users toward effective refinements.” -- benefit
Line 74: 'exisitng' -> existing.
Line 95: “…both these two works and their related subsequent works…” -- refine English.

**Quality:**

3

**Strengths And Weaknesses:**

+ Overall, this is a well-written paper that is easy to follow.
+ The authors have clearly summarized the differences from existing approaches such as ChatIR and PlugIR.
+ If the dataset is made public, it would be a valuable contribution to further research in this area.

In particular, I am summarising the strengths as follows:

(1) Instead of verbal feedback, as demonstrated by prior works like ChatIR—where a language model is used for feedback—and PlugIR—where an image captioning model is used—the authors propose using a text-to-image generation model to produce a visual feedback image that closely reflects the user’s intent. This approach offers a more interpretable means of understanding the system’s behavior through visual analysis, rather than relying solely on textual feedback as in earlier methods such as ChatIR or PlugIR.

(2) This explicit visualization makes it more interpretable, allowing the user to understand the discrepancy between the previous prompt and the intended image, and how to modulate or provide the next query so that the retrieved image aligns with the one the user has in mind.

(3) The authors have also developed a multi-round dataset, with each round consisting of a refined query, a generated synthetic image, and retrieved results with a correctness label. If this dataset is made public, it could initiate more research in this topic. More specifically, although the authors are not currently using any RL-based fine-tuning, this labeled dataset could be used to fine-tune the Image Generator or the Image-to-Image retrieval model in a customized way that is optimized in a more user-specific manner. For reference, the following paper could be considered: ‘DPOK: Reinforcement Learning for Fine-tuning Text-to-Image Diffusion Models,’ NeurIPS 2023. https://arxiv.org/abs/2305.16381.


Next, I am summarising the weaknesses that might be worth considering and addressing further.

(1) The experimental evaluation depends on the VLM-simulated users to simulate or mimic the real user interactions in Mental Image Retrieval (MIR) tasks. This means that the users always have a clear, fixed mental image. I wonder if it will be worth it, because in the real world, mental representations are always somewhat vague at the beginning and evolve with iterative feedback. Some clarification would be helpful.

(2) Many a time, diffusion-based generative models do not capture (https://arxiv.org/pdf/2403.11821) semantically aligned latent concepts of the human intent. Visual feedback may misrepresent the user’s query, especially when generation quality is poor. This will lead to misleading verbal feedback. Some analysis is needed around this.

(3) The authors have used off-the-shelf text-to-image generators like Infinity [6], Lumina-Image-2.0 [22], Stable Diffusion 3.5 [4], FLUX.1 [10], and HiDream-I1 [26] and a retrieval framework (BLIP-2).  This is more of an adaptation of existing things in the context of a visual feedback-based iterative image retrieval framework. Some discussion on how this framework could be customized in a more problem-specific manner, with specific training tricks or architectural modifications, would be beneficial. For example, RL-based fine-tuning with LoRA weights.

Overall, the novelty of this work is somewhat limited, but the self-contained writing and insightful visuals make it, in my opinion, a reasonably competitive submission. This is more of a 'novel system-level' configuration for mental image retrieval with in-depth experimental results.

---

> ### Author Rebuttal · Authors · 2025-07-31
>
> We sincerely thank the reviewer for the insightful feedback and pointing out the value of the dataset, the interpretability of the GenIR framework, and the novelty of using visual feedback to reflect user intent. We are happy to clarify the excellent questions raised.
>
> ***W1/Q6: VLM simulation and Generalization to Real Users***
>
> Our use of VLM simulation was a deliberate methodological choice for this foundational study. To scientifically validate our core hypothesis, i.e., the generative visual feedback is superior to verbal, we required a controlled, reproducible, and large-scale experimental setup. This allowed us to isolate the feedback variable and empirically demonstrate its significant impact on performance, a necessary prerequisite before undertaking complex and costly human studies. Furthermore, we expect that for users with the fuzzy mental images the reviewer describes, our visual feedback is even more critical. It provides a concrete artifact that helps the user clarify their own intent, a dynamic we are excited to explore in future HCI studies. We see our work as the first research step towards the Mental Image Retrieval task, providing the crucial proof of concept and the first high-performance, open-source testbed (codebase and dataset) for the community to now investigate nuanced questions, such as fuzzy mental models.
>
>
>
> ***W2/Q5: Failure analysis on generated images***
>
> We thank the reviewer for raising this important point. In Appendix D, we have investigated this issue through a human evaluation of visual feedback utility. Our analysis reveals that visual feedback can, in certain cases, mislead the verbal feedback by conveying incorrect or uninformative cues. As a complementary analysis to our main study, we have conducted a detailed analysis of failure modes, identifying three primary patterns where generated visual feedback can mislead the refinement process by providing incorrect cues.
>
>
> | Error Types | Description | Example (MSCOCO) |
> |-------------|-------------|------------------|
> | **Limited improvement** | The generated image shows minimal visual change from the previous round despite query refinement, providing no new information. | For image_000000004952, the generated images from the 8th and 9th rounds are nearly identical, stalling the refinement process. |
> | **Hallucination content** | The image contains objects or scene elements that were not part of the user's intent, diverting the refinement process. | For image_000000020415, the 8th-round image hallucinates a shower head on the right side of the frame, an element entirely absent in the ground-truth, which could misguide the user's next query. |
> | **Retrieval-detail misalignment** | The generated image misrepresents details that are crucial for retrieval but may be perceived as trivial by observers, creating a gap between visual feedback and retrieval objectives. | For image_000000026494, a long wooden bench in the ground-truth is incorrectly rendered as a single wooden chair in the generated image. While seemingly minor, this detail is critical for accurate retrieval but may be overlooked during refinement.
>
> Due to the restriction on uploading images, we are unable to provide the visualization. We will add the detailed analysis and visualized examples to the appendix in the final version to provide a more complete picture of the framework's current limitations and guide future improvements.
>
>
> ***W3/Q2/Q4: System training and optimization***
>
> Our current plug-and-play design is intentional: it provides a flexible and accessible framework for researchers to immediately leverage SOTA performing or in-domain customized models as they choose. We also believe this setting is handy specifically for HCI researchers who plan to conduct experiments with human subjects. That being said, model optimization aligns perfectly with our vision, thus, we share a discussion on that here.
> Freezing the retriever and optimizing the image generator in an end-to-end manner may be the best choice. On the retriever side, we expect that the retriever may not necessarily be a trainable deep learning model to provide greater flexibility, such as allowing the use of Google Search in the general domain or the use of users' own well-designed rule-based search in some specific fields. On the generator side, the success of the image generator depends on whether the generated image can "correctly reflect the text query" and "distinguish between retrieval targets and incorrect search candidates in the image space". In other words, we want the collaboration between components to be seamless.
> In this case, we highly agree with your suggestion that end-to-end reinforcement learning is necessary. However, **the data may become an obstacle, which highlights our data contribution.** As you astutely suggested, the dataset we released provides future research such opportunities for end-to-end reinforcement learning. Each of our data points contains the entire execution trace (text query -> generated image -> retrieved result). As shown in appendix A.1 and figure 6, **the quality of our data is significantly improved based on ChatIR**, which is the best resource in this direction at this moment.
>
> ***Q1: Technical contribution and novelty***
>
> We thank the reviewer for this question, as it allows us to clarify our core contribution. While the components are plug-and-play, the **novelty lies in the system-level paradigm itself**. Our key innovation is the **generate-visualize-retrieve closed loop**, which fundamentally reframes interactive search. Unlike prior work, GenIR uses the generated image not as the final output, but as an internal representation made external. This artifact serves two novel functions: (1) it provides explicit, interpretable visual feedback to the user, and (2) it acts as the query object for a more robust uni-modal (image-to-image) search. This framework, which was previously infeasible, is our main contribution.
>
> ***Q3: Cost-Benefit Analysis***
>
> We agree that latency is a key consideration. To provide a clear cost-benefit analysis, we performed an analysis to quantify the potential of a **hybrid system** that chooses between fast Verbal Feedback and our more powerful Visual Feedback.
>
> Our analysis shows that Visual Feedback **uniquely succeeds in 22.3% of cases** where Verbal Feedback fails. We present two hybrid scenarios: (1) an **"Oracle"** that perfectly selects when to use Visual Feedback, and (2) a **"Random Select"** baseline that randomly chooses Visual Feedback for 22.3% of queries.
>
> As shown in the table 1, even random selection yields a **+3.40% absolute improvement** (74.48% to 77.88%) at Round 0, while the oracle achieves **+18.92%**. For our experiment on MSCOCO, the average latency increased by just **+3 seconds per query**. This demonstrates that even a simple hybrid approach provides **meaningful gains**, with **significant room for improvement** through intelligent selection mechanisms. Furthermore, in difficult cases where verbal feedback requires many rounds (or hit max rounds due to continuous unsuccessful attempts), GenIR's ability to **succeed in fewer rounds** can actually be more time-efficient overall. We will add this full analysis to the final paper.
>
> ### Table 1: Comparison of using Verbal Feedback, Visual Feedback, and Hybrid Systems for Image Retrieval (Hits@10)
> *22.3% Visual / 77.7% Verbal*
> | Dialog Length | Verbal Feedback | Visual Feedback | Hybrid Oracle | Random Select | Verbal->Oracle | Verbal->Random |
> |--------------|-----------------|-----------------|---------------|---------------|----------------|----------------|
> | 0            | 74.48%         | 89.71%         | 93.40%        | 77.88%        | +18.92%        | +3.40%         |
> | 1            | 82.73%         | 93.11%         | 95.97%        | 85.06%        | +13.25%        | +2.33%         |
> | 2            | 85.83%         | 95.00%         | 97.43%        | 87.89%        | +11.60%        | +2.06%         |
> | 3            | 87.97%         | 95.97%         | 97.91%        | 89.76%        | +9.95%         | +1.79%         |
> | 4            | 89.13%         | 96.51%         | 98.20%        | 90.81%        | +9.07%         | +1.68%         |
> | 5            | 89.96%         | 96.85%         | 98.30%        | 91.50%        | +8.35%         | +1.54%         |
> | 6            | 90.49%         | 97.14%         | 98.59%        | 91.98%        | +8.10%         | +1.49%         |
> | 7            | 90.98%         | 97.48%         | 98.64%        | 92.42%        | +7.67%         | +1.44%         |
> | 8            | 91.27%         | 97.67%         | 98.79%        | 92.69%        | +7.52%         | +1.42%         |
> | 9            | 91.80%         | 97.72%         | 98.79%        | 93.13%        | +6.99%         | +1.33%         |
> | 10           | 92.33%         | 98.01%         | 98.88%        | 93.62%        | +6.55%         | +1.29%         |

---

> > ### Comment · Reviewer_bt51 · 2025-08-06
> > **Discussion**
> >
> > Is it possible to suggest how RL-based finetuning could be used on top of this framework as a future work discussion?

---

> > > ### Author Response · Authors · 2025-08-07
> > >
> > > We thank the reviewer for encouraging this forward-looking discussion. While a deep dive into specific optimization algorithms is beyond the scope of this work, we are excited about the future possibilities our work enables. We view these opportunities from two perspectives: the framework itself and the unique dataset it produces.
> > >
> > > **A GenIR Framework**
> > >
> > > At its core, the GenIR framework acts as a multi-agent system where the generator and retriever collaborate to solve the user's retrieval task. By introducing an explicit *visual feedback* channel into this loop, we create a new interactive paradigm that is potentially *amenable to established multi-agent optimization techniques*. Meanwhile, its modular design makes GenIR a flexible testbed. Researchers can plug-and-play different agents (e.g., swapping a diffusion model for a GAN, or a neural retriever for a rule-based one) to investigate how agent capabilities impact system dynamics and the potential for optimization.
> > >
> > > **A Trajectory-rich Dataset**
> > >
> > > Our dataset is a collection of rich, multi-step interaction trajectories that supports further system optimization and analysis:
> > >
> > > * **Warm-up:** The successful (*query -> generated image -> retrieval result*) examples in our dataset can be used for an initial "warm-up" stage before RL. This makes the policy network (large models) stronger, which is a standard practice for improving the stability and sample efficiency of the subsequent RL stage. This is particularly crucial for navigating the vast action space inherent to large generative models, guiding exploration and accelerating convergence.
> > >
> > > * **Trajectory Optimization:** The dataset's full interaction histories are crucial for advanced trajectory-based optimization. They enable *process supervision*, e.g., by training a reward model, instead of solely relying on final, delayed reward.
> > >
> > > * **Diagnostic Analysis:** The intermediate images offer an unprecedented resource for analyzing *how* and *why* generative models misunderstand queries in a task-oriented setting. This provides direct, actionable guidance for system-level debugging and task-specific optimization.
> > >
> > > Note that the possibilities we have outlined above are intended as only a starting point to engage the research discussion around this big topic (Optimization on a multi-agent system), and we believe that the community will discover many other opportunities not covered here based on this work.
> > > In essence, our work provides the essential building blocks, i.e., a **flexible framework** and a **trajectory-rich dataset**, for the community. By doing so, we hope to pave the way for research that unlocks the full potential of **generative visual feedback** in interactive AI tasks.

---

> ### Comment · Area_Chair_1c4L · 2025-08-05
>
> Reviewer bt51, please engage in the discussion period. I understand that the review period started over a weekend, but we only have a few days remaining in the (now slightly extended) discussion period. The authors have provided a thoughtful response to your review and you are obligated to respond to it. You should share with the authors if they addressed questions or concerns you had, and seek clarification about any questions or concerns that remain. Please post your response as soon as you can so that there is time for the authors to follow up and discussion to progress as needed.

---

### Official Review · Reviewer_6Fz4 · 2025-07-02

**Clarity:** 3
**Significance:** 3
**Originality:** 3
**Rating:** 5
**Confidence:** 3

**Summary:**

This paper introduces Mental Image Retrieval (MIR) as a novel task addressing the gap between standard text-to-image retrieval and real-world search behavior, where users iteratively refine queries based on a mental image. The authors propose GenIR, a multi-round retrieval framework leveraging text-to-image diffusion models to generate synthetic visual feedback at each interaction step. This synthetic image explicitly reveals the system’s interpretation of the user’s query, enabling intuitive refinement. Experiments show significant gains over verbal-feedback baselines (e.g., +15–20% Hits@10 on FFHQ/Clothing-ADC) across four datasets, with robustness to generator quality and model scale.

**Questions:**

1. Conduct larger-scale human studies (≥50 users) comparing GenIR vs. verbal feedback for tasks with fuzzy mental images (e.g., "find a scenic mountain photo I saw last year"). If human studies show statistically significant improvements (e.g., +20% success rate in ≤5 rounds), this strengthens real-world applicability. If results are marginal, significance claims may weaken.
2. Analyze cases where generated feedback misleads refinement (e.g., Appendix’s "unhelpful" cases). Quantify error types (e.g., hallucinated objects, incorrect attributes).
3. Compare GenIR against a hybrid baseline (e.g., generate images only when verbal confidence is low) or discuss optimizations (e.g., distillation to faster generators).

**Ethical Concerns:**

["NO or VERY MINOR ethics concerns only"]

**Final Justification:**

Satisfied that the majority of my concerns have been resolved, I vote to accept.

**Limitations:**

yes

**Quality:**

3

**Strengths And Weaknesses:**

# Strengths
1. Addresses a critical limitation of VLMs in real-world search settings. The task definition, strong results on 1M+ image datasets (Clothing-ADC), and model-agnostic framework provide foundations for future work.
2. The integration of generative models for visual feedback in interactive retrieval is new. Prior work (e.g., ChatIR, PlugIR) relies on textual dialogue, while GenIR’s image-based feedback offers a tangible advantage for disambiguation. Rigorous experiments across diverse domains (COCO, FFHQ, Flickr30k, Clothing-ADC), ablation studies (diffusion models, VLM sizes), and automated dataset validation (Table 1) support claims.

# Weaknesses
1. Reliance on VLM simulations (Gemma3) instead of human users weakens ecological validity. Human evaluation (Appendix D) is limited (100 samples, single annotator).
2. Computational overhead of iterative image generation (Table 3) is under-discussed. Latency (16–27s/round) may limit real-world deployability.
3. While the framework is novel, components (diffusion models, image retrieval) are off-the-shelf. The claim of "first generative feedback for retrieval" may be accurate but could be more nuanced.

---

> ### Author Rebuttal · Authors · 2025-07-31
>
> We thank the reviewer for their valuable feedback and for acknowledging the significance of the MIR task, the novelty of our framework, and the rigor of our experiments. We appreciate the opportunity to address the important points raised.
>
> ***W1/Q1: Size of human evaluation and future larger-scale human studies.***
>
> We agree that large-scale human studies are the gold standard for evaluating interactive systems. However, our use of VLM simulation was a deliberate and necessary methodological choice for this foundational work, intended to establish the core technical validity of our proposed paradigm.
>
> **Our primary scientific objective was to rigorously test the hypothesis that generative visual feedback is fundamentally superior to verbal feedback.** To do this, we needed to isolate the impact of the feedback mechanism itself. A controlled, reproducible simulation was scientifically necessary to perform the large-scale quantitative analysis (across 4 datasets, including one with 1M+ images) required to validate this hypothesis. This approach, common in pioneering new interactive tasks (e.g., ChatIR [12], PlugIR [11]), allowed us to robustly demonstrate significant performance gains that would be impossible to establish in a preliminary human study.
>
> Our work serves as the crucial proof of concept that modern generative models are now capable of supporting this new retrieval paradigm. We see our framework and dataset not as the final word, but as **the first research step towards Mental Image Retrieval**. By providing a high-performing, open-source testbed (codebase and dataset), we have lowered the barrier to entry, enabling the broader range of AI community beyond ML researchers, e.g., HCI and HAI experts, to now investigate the more nuanced questions of fuzzy mental models and evolving user intent that were previously intractable. This controlled validation was a necessary prerequisite to justify the significant investment of complex, costly human-in-the-loop studies, which is the clear and exciting next phase of this research program. Our preliminary human evaluation in Appendix D, which found 86% of the visual feedback to be useful, serves as an initial validation that corroborates our main findings.
>
> *[11] Lee, Saehyung, et al. "Interactive Text-to-Image Retrieval with Large Language Models: A Plug-and-Play Approach." ACL. 2024.*
>
> *[12] Levy, Matan, et al. "Chatting makes perfect: Chat-based image retrieval." NeurIPS 2023.*
>
> ***W2/Q3: Latency concern and future optimization***
>
> We agree that latency is a critical factor for real-world deployment. While our paper does not claim a contribution in efficiency, our plug-and-play framework is designed to readily incorporate state-of-the-art techniques from the Efficient Generative AI community, such as faster diffusion models[3], faster computing[4], as well as future advances.
>
> To directly illustrate the cost-benefit trade-off, we performed an analysis to quantify the potential of a **hybrid system** that chooses between fast Verbal Feedback and our more powerful Visual Feedback.
>
> Our analysis shows that Visual Feedback **uniquely succeeds in 22.3% of cases** where Verbal Feedback fails. We present two hybrid scenarios: (1) an **"Oracle"** that perfectly selects when to use Visual Feedback, and (2) a **"Random Select"** baseline that randomly chooses Visual Feedback for 22.3% of queries.
>
> As shown in the table 1, even random selection yields a **+3.40% absolute improvement** (74.48% to 77.88%) at Round 0, while the oracle achieves **+18.92%**. For our experiment on MSCOCO, the average latency increased by just **+3 seconds per query**. This demonstrates that even a simple hybrid approach provides **meaningful gains**, with **significant room for improvement** through intelligent selection mechanisms. Furthermore, in difficult cases where verbal feedback requires many rounds (or hit max rounds due to continuous unsuccessful attempts), GenIR's ability to **succeed in fewer rounds** can actually be more time-efficient overall. We will add this full analysis to the final paper.
>
>
> ### Table 1: Comparison of using Verbal Feedback, Visual Feedback, and Hybrid Systems for Image Retrieval (Hits@10)
>
> *22.3% Visual / 77.7% Verbal*
> | Dialog Length | Verbal Feedback | Visual Feedback | Hybrid Oracle | Random Select | Verbal->Oracle | Verbal->Random |
> |--------------|-----------------|-----------------|---------------|---------------|----------------|----------------|
> | 0            | 74.48%         | 89.71%         | 93.40%        | 77.88%        | +18.92%        | +3.40%         |
> | 1            | 82.73%         | 93.11%         | 95.97%        | 85.06%        | +13.25%        | +2.33%         |
> | 2            | 85.83%         | 95.00%         | 97.43%        | 87.89%        | +11.60%        | +2.06%         |
> | 3            | 87.97%         | 95.97%         | 97.91%        | 89.76%        | +9.95%         | +1.79%         |
> | 4            | 89.13%         | 96.51%         | 98.20%        | 90.81%        | +9.07%         | +1.68%         |
> | 5            | 89.96%         | 96.85%         | 98.30%        | 91.50%        | +8.35%         | +1.54%         |
> | 6            | 90.49%         | 97.14%         | 98.59%        | 91.98%        | +8.10%         | +1.49%         |
> | 7            | 90.98%         | 97.48%         | 98.64%        | 92.42%        | +7.67%         | +1.44%         |
> | 8            | 91.27%         | 97.67%         | 98.79%        | 92.69%        | +7.52%         | +1.42%         |
> | 9            | 91.80%         | 97.72%         | 98.79%        | 93.13%        | +6.99%         | +1.33%         |
> | 10           | 92.33%         | 98.01%         | 98.88%        | 93.62%        | +6.55%         | +1.29%         |
>
>
> *[3] Chen, Junsong, et al. "Sana-sprint: One-step diffusion with continuous-time consistency distillation." arXiv preprint arXiv:2503.09641 (2025).*
>
> *[4] Wu, Chengyue, et al. "Fast-dllm: Training-free acceleration of diffusion llm by enabling kv cache and parallel decoding." arXiv preprint arXiv:2505.22618 (2025).*
>
> ***W3: Claimed novelty***
>
> We thank the reviewer for affirming the framework's novelty. We agree on the nuance of meaning within our claim. Meanwhile, we believe that our claim in line 114, i.e., “... **the first work to integrate text-to-image generation into an interactive retrieval setting, enabling a closed-loop interaction that unifies generation, retrieval, and feedback within a single framework.**” is accurate. Our core novelty is not merely the use of a generator, but the proposal of a new interactive paradigm and the system-level framework (GenIR) that realizes it. The originality lies in the generate-visualize-retrieve closed loop, which fundamentally changes user-system interaction by making the system's internal understanding explicit and actionable.
>
> ***Q2: Analyze on failure examples***
>
> This is an excellent suggestion. We have conducted a focused analysis of failure modes where generated feedback can mislead the refinement process. These errors primarily occur when the generated image provides incorrect visual cues. We have categorized them as follows:
>
> | Error Types | Description | Example (MSCOCO) |
> |-------------|-------------|------------------|
> | **Limited improvement** | The generated image shows minimal visual change from the previous round despite query refinement, providing no new information. | For image_000000004952, the generated images from the 8th and 9th rounds are nearly identical, stalling the refinement process. |
> | **Hallucination content** | The image contains objects or scene elements that were not part of the user's intent, diverting the refinement process. | For image_000000020415, the 8th-round image hallucinates a shower head on the right side of the frame, an element entirely absent in the ground-truth, which could misguide the user's next query. |
> | **Retrieval-detail misalignment** | The generated image misrepresents details that are crucial for retrieval but may be perceived as trivial by observers, creating a gap between visual feedback and retrieval objectives. | For image_000000026494, a long wooden bench in the ground-truth is incorrectly rendered as a single wooden chair in the generated image. While seemingly minor, this detail is critical for accurate retrieval but may be overlooked during refinement. |
>
> Due to the restriction on uploading images, we are unable to provide the visualization. We will add the detailed analysis and visualized examples to the appendix in the final version to provide a more complete picture of the framework's current limitations and guide future improvements.

---

> ### Comment · Area_Chair_1c4L · 2025-08-05
>
> Reviewer 6Fz4, please engage in the discussion period. I understand that the review period started over a weekend, but we only have a few days remaining in the (now slightly extended) discussion period. The authors have provided a thoughtful response to your review and you are obligated to respond to it. You should share with the authors if they addressed questions or concerns you had, and seek clarification about any questions or concerns that remain. Please post your response as soon as you can so that there is time for the authors to follow up and discussion to progress as needed.

---

### Note · Authors · 2025-08-12

Dear Reviewers, ACs, and SACs,

We appreciate your time and thoughtful feedback, which we believe will further enhance the quality of our work.

--------------------------------------------------------------------------------------------------------------------------
Strengths highlighted by the reviewers:
- **Clear motivation & practical relevance [6Fz4, bt51, TmpY]**. The paper addresses a critical, real-world motivated problem with an interpretable approach.
- **Innovative, paradigm-shifting framework [6Fz4, bt51, KHQJ]**. GenIR is recognized as a novel and original framework that reframes interactive search through its *generate-visualize-retrieve* closed loop.
- **Valuable dataset [bt51]**. The released dataset is a valuable contribution that provides a high-quality resource for future research in the community.
- **Extensive, convincing experiments [6Fz4, KHQJ]**. The rigorous experiments across four diverse datasets, including one with a million-scale image database, demonstrate strong performance of the proposed framework.
---
We have also clarified the following points, which we will make clear in the final version
- **Novelty clarification [bt51, TmpY]**. We elaborated that our core novelty lies in the system-level, closed-loop paradigm.
- **Research scope justification [6Fz4, bt51, TmpY]**. We clarified that while this work does not aim to make a direct impact on the HCI field through human studies, it serves as the crucial first research step towards the Mental Image Retrieval task.
- **Contribution of strong initial performance [TmpY]**. We explained that the strong performance at the 0th iteration validates our superior core retrieval mechanism, while subsequent gains validate the interactive feedback loop.
- **Future research on RL optimization [bt51]**. Thanks to the reviewer's forward-looking question, we highlighted how our trajectory-rich dataset and modular framework provide invaluable opportunities for future research.
---
We appreciate the reviewers’ detailed suggestions, which we have addressed during the rebuttal phase and will incorporate into the final version:
- **Computational cost & efficiency [6Fz4, bt51, KHQJ]**. We conducted a cost-benefit analysis of a hybrid system, demonstrating that significant performance gains are achievable with minor latency overhead.
- **Failure mode analysis [6Fz4, bt51]**. We performed a focused failure mode analysis, categorizing them into three categories.
---

---

### Decision · Program_Chairs · 2025-09-17

**Decision:**

Accept (poster)

**Comment:**

The paper proposes GenIR, a generative framework for a new interactive search paradigm — multi-round mental image retrieval, where users iteratively refine queries based on visual feedback generated by diffusion models.

Reviewers thought this new retrieval paradigm was very interesting (and the AC concurs), with a compelling motivation and a clear system-level innovation. They thought that GenIR offers an intuitive and interpretable user experience through visual feedback, and noted that it outperforms verbal-feedback-based baselines across multiple datasets. Reviewers also appreciated the released dataset, believing it would be a valuable resource for future work.

Some weaknesses were noted. Chief among them was the reliance on simulated VLM-based users rather than human evaluations, raising questions about ecological validity. Other concerns included the computational cost of generating visual feedback and the use of off-the-shelf components. However, it seems that reviewers were largely satisfied by the authors responses to the concerns during the rebuttal and discussion.

One reviewer initially leaned toward a borderline reject, questioning the novelty and iterative nature of the framework due to strong performance at iteration 0. However, the authors clarified that this performance reflects a strong base retriever, and the additional gains across rounds demonstrate the utility of the feedback loop. The reviewer acknowledged these clarifications and ultimately updated their stance (although their actual rating does not appear to have been updated, the AC interprets their final justification as suggesting a borderline accept score).

For the camera-ready version, the authors should include the expanded failure analysis, visualized examples, and hybrid system evaluation that were promised in the discussion and rebuttal.

The AC recommends acceptance.